# Stochastic optimization for minimizing operational costs in smart hybrid energy networks considering electric vehicle

**Nouman Qamar**[1], **Mohammed Alqahtani**[2], **Muhammad Rehan**[3,4], **Ijaz Ahmed**[4]*, **Muhammad Khalid**[3,4]*

**1** Electrical Engineering Department, University of Engineering and Technology, Punjab, Pakistan, **2** Department of Industrial Engineering, King Khalid University, Abha, Saudi Arabia, **3** Electrical Engineering Department, King Fahd University of Petroleum & Minerals (KFUPM), Dhahran, Saudi Arabia, **4** Interdisciplinary Research Center for Sustainable Energy Systems (IRC-SES), KFUPM, Dhahran 31261, Saudi Arabia

* ijaz.ahmed.1@kfupm.edu.sa (IA); mkhalid@kfupm.edu.sa (MK)

**Data availability statement:** All relevant data are within the manuscript and in supporting information file.

## Abstract

The residential energy hub (REH) effectively satisfies power demands, but the incorporation of renewable energy sources (RES) and the increasing use of plug-in hybrid electric vehicles (PHEVs), with their unpredictable nature, complicates its optimal functionality and challenges the accurate modeling and optimization of REH. This work proposed a stochastic model for REH using mixed integer linear programming (MILP) to optimally handle the associated uncertainties of RES and PEHVs, which was then solved using GAMS software. Four case studies with varying conditions were conducted to verify the performance of the proposed scheme, and the results indicate that the approach is superior in optimally handling the system's associated limitations. These limitations include the intermittency and variability of RES and the uncertainties associated with PHEVs, such as arrival time, travel distance, and departure time. Additionally, this work introduces a smart charging mechanism that charges and discharges PHEVs economically, both in terms of cost and reliability. The results indicate that incorporating a smart charging mechanism decreases the total operating cost of smart REH by 2.59% while maintaining the comfort level of the consumer and increasing the reliability of the overall system. Finally, smart REH adopts a demand response program (DRP), which further reduces the operational cost by 3.7%. Furthermore, the proposed approach demonstrates a significant reduction in operating costs and an improvement in the reliability of the smart REH.

## 1 Introduction

The global energy demand for the residential sector is increasing rapidly. Anticipated growth in residential sector demand is likely to result in more reliance on renewable energy sources (RES) and novel energy storage systems (ESSs) [1,2]. Home energy management systems

**Funding:** The authors would like to acknowledge the financial support provided by King Abdullah City for Atomic and Renewable Energy (K.A.CARE) in conducting this research at KFUPM. Additionally, author Mohammed Alqahtani (M.A) acknowledges the support of the Deanship of Scientific Research at King Khalid University, Saudi Arabia, under the Large Group Research Project RGP2/392/45.

**Competing interests:** The authors have declared that no competing interests exist.

(HEMS) now simultaneously meet the electrical, cooling, and heating requirements of residential buildings due to the emergence of the energy hub (EH) concept and improved energy form conversion efficiency [3,4]. Residential facilities primarily consume energy due to their electrical, heating, and cooling requirements. However, optimizing the functioning of these systems presents a challenge, as it is crucial for enhancing the overall efficiency of both electricity and gas grids [5,6]. This highlights the critical research gap in the development of advanced solutions to improve the integration and control of energy resources in residential buildings [7,8].

The EH concept, which can handle and convert different forms of energy, is gaining popularity in research for the optimization of HEMS [9,10]. Adopting different energy management techniques from EH can optimize the energy purchasing cost of residential buildings [11]. Residential buildings are increasingly using RES to reduce energy costs and reduce their reliance on fossil fuels [12]. It is crucial to integrate RES into EH systems to enhance their contribution to the building's energy consumption. However, given their intermittent nature, accurately predicting and optimizing their share in meeting the demands of REH is a challenging task [13]. The use of ESS can mitigate the uncertainty associated with the RES, increasing system reliability and lowering operational costs [14–16]. The use of an electric vehicle (EV) in place of a conventional vehicle is increasing rapidly for a cleaner environment. The use of EVs in residential buildings increases the burden on the electrical network, but proper utilization of EV batteries can improve energy cost optimization [17,18]. The availability of EVs in EH is dependent on the unpredictable departure and arrival times of the EV's owner [19]. This limits the use of EVs as a complete storage device for energy. Additionally, EVs require smart charging for their daily travel routine, as improper charging could burden the REH and reduce its efficiency [20,21]. The proper size and smart charging are required to properly and optimally integrate the EV with the REH [22,23].

Despite extensive research on EV integration in EH, it lacks the complete stochastic modeling of all the uncertain components necessary for precise and accurate results in the shortest amount of time [24,25]. Therefore, there is a pressing need to propose a fast and accurate technique for the optimal operation of REH with RES and PHEV [26,27]. Developing a new technique based on scenario generation and reduction methods, in conjunction with the smart charging of PHEVs, would provide a suitable solution for the optimal operation of REH, ensuring speed and accuracy. The uncertainties of RES and PHEV availability pose an adverse effect on the operation of REH, but by identifying and negating their negative impact, optimal operation of REH can be achieved [28,29]. Scholars in [30], optimal energy scheduling of REH in the presence of storages is discussed along with demand side management (DSM). A two-stage multi-criteria optimization technique is presented that improves the storage lifetime of storage devices in the first step and optimizes the REH in the second step [31]. Although this technique increases the life of energy storage devices, it lacks the inclusion of intermittent RES and uncertain PHEV. The study [32] proposes a solution to the scheduling problem of smart REH (SREH), which considers various uncertain parameters such as solar radiation, energy demands, and electricity costs, using the Monte Carlo simulation technique. However, it ignores the integration of PHEV into REH. In [33], authors introduced an IoT-enabled approach for optimizing multi-EH, including RES and PHEV, that addresses the uncertain nature of RES in a correlated environment. In [34], we present a deep learning-based optimal scheduling of a virtual energy hub in the presence of plug-in hybrid CNG (compressed natural gas) EVs.

Some references have focused on the handling of uncertainties associated with RES through probabilistic or stochastic modeling [35]. The authors in [36] developed a probabilistic operation strategy for the optimal operation of EH with different energy converters

by proposing various innovative uncertainty modeling techniques. In [37], the authors propose a probabilistic correlation of RES in EH to maximize its profit, based on the game theory approach. In study [38], the authors discuss a stochastic framework for energy management for SEH that utilizes a false data detection scheme. Wang et al. [39], presents a new stochastic optimal SREH management system that seeks to reduce energy usage, GHG emissions, and energy expenditures by determining the optimal power output for desert regions. Abdul et al. [40], researchers investigate a two-stage stochastic optimization of EH that considers different types of storage in the presence of RES and EVs. The first stage focuses on determining the optimum size and placement of each EH, while the second stage addresses the optimum charging and discharging of each component and EV. The results show that smart charging and discharging of EVs can reduce the total operating cost by approximately 5.5%. Some authors have applied the concept of a microgrid as a decentralized EH by integrating and managing different energy sources. Their dynamic model of control systems enables efficient, sustainable, and resilient energy management, allowing it to operate independently from the main grid while providing reliable power [41,42]. Shokri et al. [43], introduced a novel concept for the optimal operation of smart cities as energy hubs. Their EH model incorporates heat pumps, fuel cells, microturbines, ESSs, desalination units, RES, and EVs. They applied a biogeography-based optimization algorithm to an enhanced IEEE 33-bus test system to demonstrate its effectiveness. The results show a significant decrease in costs and environmental pollution.

Many scholars have successfully integrated the RES to REH, and some have also effectively managed the uncertainty. Table 1 reveals the major novelty of proposed work in terms of modeling and associated constraints and research gap in the stochastic modeling and integration of PHEV, along with its smart charging and DSM. While some researchers have successfully integrated the PHEV into the REH and addressed its uncertainty, the absence of a

**Table 1. Comparison of the proposed SREH with relevant literature.**

| References | Uncertainty modeling of RES | DSM | PHEV | Uncertainty modeling of PHEV | Smart charging of PHEV | |
|---|---|---|---|---|---|---|
| [44] | ✓ | ✓ | ✓ | × | × | × |
| [45] | ✓ | ✓ | ✓ | × | × | × |
| [40] | ✓ | ✓ | ✓ | ✓ | × | × |
| [46] | ✓ | ✓ | × | × | × | × |
| [47] | ✓ | ✓ | ✓ | × | × | × |
| [48] | ✓ | ✓ | ✓ | ✓ | × | × |
| [49] | ✓ | ✓ | × | ✓ | ✓ | × |
| [50] | ✓ | ✓ | ✓ | × | × | × |
| [51] | ✓ | ✓ | ✓ | ✓ | × | × |
| [52] | ✓ | × | × | ✓ | ✓ | ✓ |
| [53] | ✓ | × | × | ✓ | ✓ | ✓ |
| [54] | ✓ | ✓ | × | ✓ | ✓ | × |
| [55] | ✓ | ✓ | ✓ | ✓ | × | × |
| [56] | ✓ | × | × | ✓ | × | × |
| [57] | ✓ | ✓ | ✓ | ✓ | × | × |
| [58] | ✓ | ✓ | ✓ | × | × | × |
| [59] | ✓ | ✓ | ✓ | ✓ | ✓ | × |
| [60] | ✓ | ✓ | ✓ | × | × | × |
| Proposed Model | ✓ | ✓ | ✓ | ✓ | ✓ | ✓ |

smart charging mechanism hinders its full optimization. Therefore, a comprehensive modeling of REH is necessary, one that not only addresses the uncertain behavior of RES and PHEV parameters, but also incorporates a smart charging/discharging mechanism for the PHEV in conjunction with the DSM system.

## 1.1 Contribution and novelties

The following outline highlights the key contributions of this research to the field of optimal stochastic operation of REH.

1. In contrast to earlier research in [61–64], a more complex and practical model has been developed that takes into account the uncertain charging limits of both RES and PHEVs, as well as factors like arrival time, travel distance, smart charging/discharging, and departure time. The developed model is more reliable and achieved optimal economic aspects in terms of cost and reliability.
2. The proposed approach signifies a notable improvement over prior research works by concurrently tackling the uncertainties linked to RES and PHEVs, resulting in enhanced accuracy and reliability of outcomes. The simultaneous examination of uncertainty improves the model's robustness and complexity, distinguishing it from previous models that addressed either RES [40,56] or PHEV challenges [52,65] individually.
3. The suggested methodology is novel relative to previous works [17,23,66], since it incorporates an intelligent decision-making process for the charging and discharging of PHEVs into the REH scheduling to enhance its optimal performance.
4. This study presents a broader strategy by concurrently applying DSM approaches to electrical, heating, and cooling loads, in contrast to prior studies such as [67,68], which concentrated exclusively on dispatch models for electric loads. The incorporation of heating and cooling systems substantially improves the model's relevance, making it more adept at handling the intricate and varied load requirements in contemporary energy systems.
5. The proposed work conducted a comparative study of all contributions (see Table 1 references [32-48]) and [69] to ascertain the relative efficacy of each comparative feature (DSM, smart charging of PHEVs, and the unpredictable nature of RES).
6. Unlike the works in [9,70–72], which use distributed methods to combine RES through RETScreen, the proposed framework is based on MILP and GAMS software, which gives better outcomes and is simple to use for complex energy delivery problems.

The proposed method introduces significant improvements over existing methods, such as smart charging strategies for PHEVs that dynamically adjust based on real-time demand, electricity prices, and the availability of RES. This feature enhances the efficiency of the SREH by optimizing the charging/discharging cycles of PHEVs in addition to the DSM. The combined stochastic modeling of solar irradiance, wind speed, and PHEV usage patterns provides more accurate forecasting and optimization under real-world uncertainties, which is crucial for optimizing EHs in smart grids [73]. Another novel aspect of our technique is the identification of the feasible operating region for the SREH components based on their constraints. Previous methods mainly focus on the isolated optimization of PHEV charging/discharging and RES without considering real-time interactions between them. The proposed method

improves on this approach by simultaneously optimizing the charging/discharging behavior of PHEVs and the energy management of the REH while also incorporating DSM strategies. This results in a more comprehensive and adaptive solution for energy management in modern grids.

The article continues with the following sections: Section 2 describes the problem description. Section 3 formulates the mathematical modeling of the problem. Section 4 describes the methodology to solve the problem. Section 5 presents simulation and case studies. The results are discussed in section 6. Finally, conclusions are drawn in section 7.

## 2 Problem description

### 2.1 REH architecture

The concept of EH has evolved to serve as an interface between diverse input energy carriers and output requirements, as noted by Najafi et al. [74]. The EH facilitates the conversion of one kind of energy into another as needed to satisfy specific demands. The fundamental model of EH is depicted in Fig. 1, with inputs denoted as I and outputs denoted as O. The inputs and outputs are connected by (1).

$$I = C * O \tag{1}$$

where $C$ represents the coupling factor between the inputs and outputs. The multiple inputs and outputs of EH make $C$ a matrix, as defined in (2) [75], which denotes the topology, dispatch factor, and converter efficiency of EH.

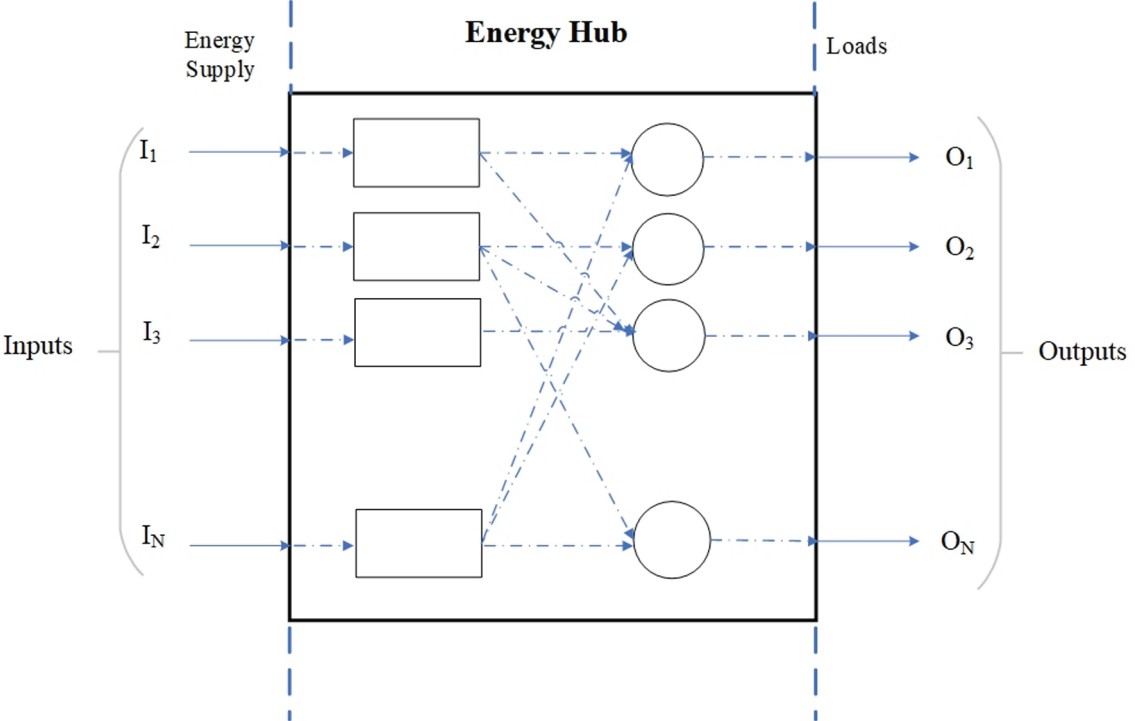

**Fig 1. Energy hub architecture.**

$$
\begin{bmatrix} I_1 \\ I_2 \\ \vdots \\ I_i \\ \vdots \\ I_N \end{bmatrix} = \begin{bmatrix} C_{11} & C_{12} & \cdots & C_{1N} \\ C_{21} & C_{22} & \cdots & C_{2N} \\ \vdots & \vdots & \ddots & \vdots \\ C_{i1} & C_{i2} & \cdots & C_{iN} \\ \vdots & \vdots & \ddots & \vdots \\ C_{N1} & C_{N2} & \cdots & C_{NN} \end{bmatrix} \begin{bmatrix} O_1 \\ O_2 \\ \vdots \\ O_i \\ \vdots \\ O_N \end{bmatrix} \tag{2}
$$

The idea of EH can be used on different levels, like residential EH, commercial EH, industrial EH, agricultural EH, etc. [76]. This paper applies the concept of EH to REH, which not only incorporates the basic components of EH but also integrates PHEV. The basic architecture of REH is shown in Fig. 2. This work incorporate a smart features such as DSM, smart charging of PHEV, accurate modeling of RES, and demands to make REH smarter such as SREH.

## 2.2 Handling procedure for uncertainties

The handling of uncertainties related to customer-owned RES and demands of REH is a challenge for the optimum scheduling of SREH [77]. Also, the uncertain behavior of PHEV owners, like time of departure and time of arrival, etc., must be properly handled to accurately model the charging/discharging of PHEV [78]. The optimization of REH is not possible without handling these uncertainties. Deterministic modeling cannot adequately handle the uncertainties related to solar irradiance and wind speed for solar and wind power production. To handle these uncertainties, either probabilistic or stochastic modeling is required, [79,80]. In this work, probability density functions (PDFs) have been applied in order to consider the random behavior of PHEVs, wind speed, and solar irradiance. The normal PDF is used to model uncertainties related to PHEVs, the Weibull PDF is used for wind speed, and the beta PDF is used to model solar irradiance. The powers generated from solar and wind are

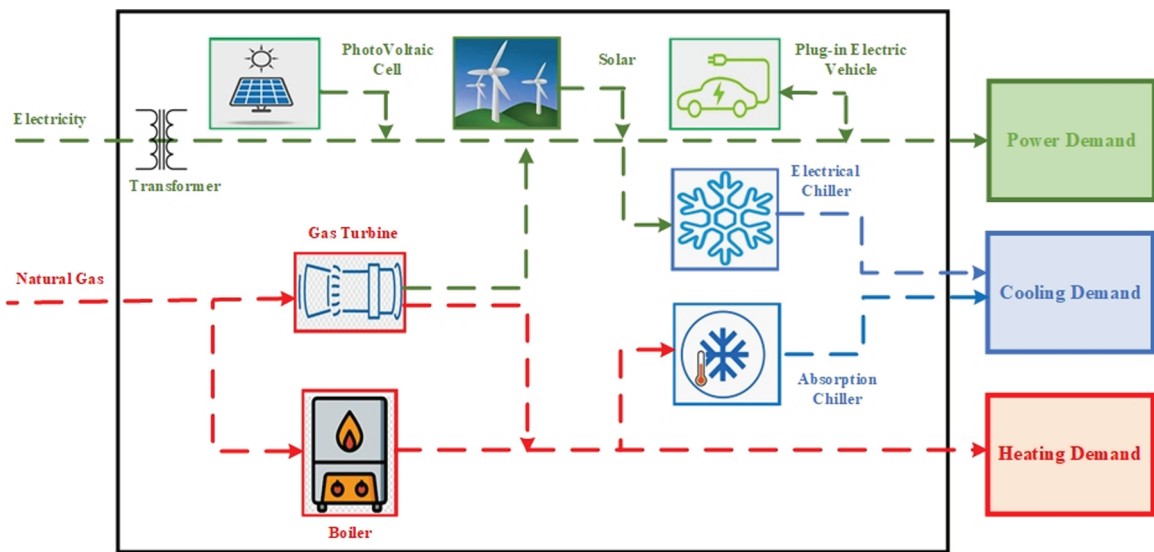

**Fig 2. Conceptual framework of REH under study.**

calculated using (3) and (4), respectively.

$$P^e_{\text{PV}}(S_i(t,s)) = \text{PV}_{\text{array}} \cdot \text{df}_{\text{PV}} \cdot S_{i,\text{STC}} \cdot \left(1 + \alpha_p \left(T_c - T_{c,\text{STC}}\right)\right) \tag{3}$$

where $PV_{\text{array}}$ is the size of PV panels, $df_{\text{PV}}$ is the derating factor of PV panels, $S_{i,\text{STC}}$ is the solar irradiance at standard test conditions, $\alpha_p$ is the coefficient of temperature, and $T_{c,\text{STC}}$ is the PV cell temperature under standard test conditions.

$$P^e_{\text{wind}}(\omega(t,s)) = \begin{cases} P^w_{\text{out}} & \text{if } \omega^w_r \leq \omega_t \leq \omega_{\text{ocut}} \\ \frac{P^w_{\text{out}}(\omega_t - \omega^{\text{cut}}_i)}{(\omega^w_r - \omega^{\text{cut}}_i)} & \text{if } \omega^{\text{cut}}_i \leq \omega_t \leq \omega^w_r \\ 0 & \text{if } \omega_t \leq \omega^{\text{cut}}_i \text{ or } \omega_t \geq \omega_{\text{ocut}} \end{cases} \tag{4}$$

where $P^w_{\text{out}}$ is the power output of wind turbine (WT), $\omega^w_r$ is the rated speed of WT, $\omega^{\text{cut}}_i$ is the cut-in speed of WT, and $\omega^{\text{cut}}_o$ is the cut-out speed of WT.

The hourly power generation of solar and wind is shown in Fig. 3.

This work handled the uncertainties related to PHEV, such as the time of arrival, departure, and traveling distance, in addition to the uncertainties related to RES. The statistical historical data of PHEV drivers is shown in Fig. 4 [81]. The mean departure time is 7:35 with a standard deviation (SD) of 0.6 hours, while the mean arrival time is 16:40 with an SD of 0.9 hours.

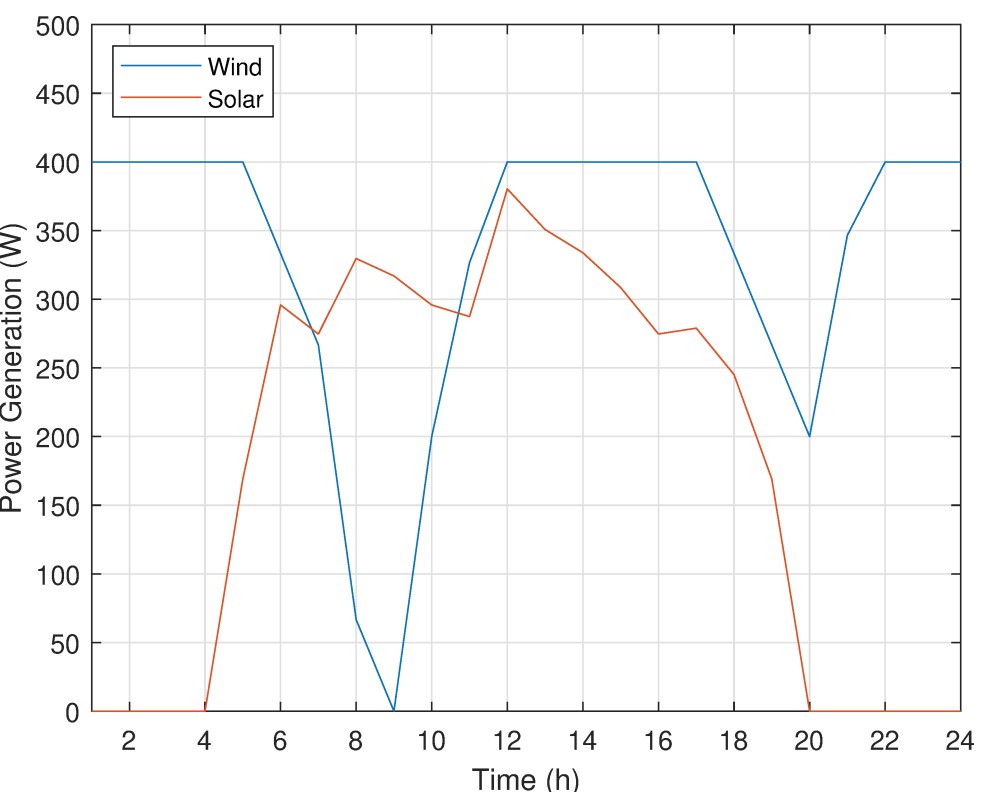

**Fig 3. Hourly power generation of solar and wind.**

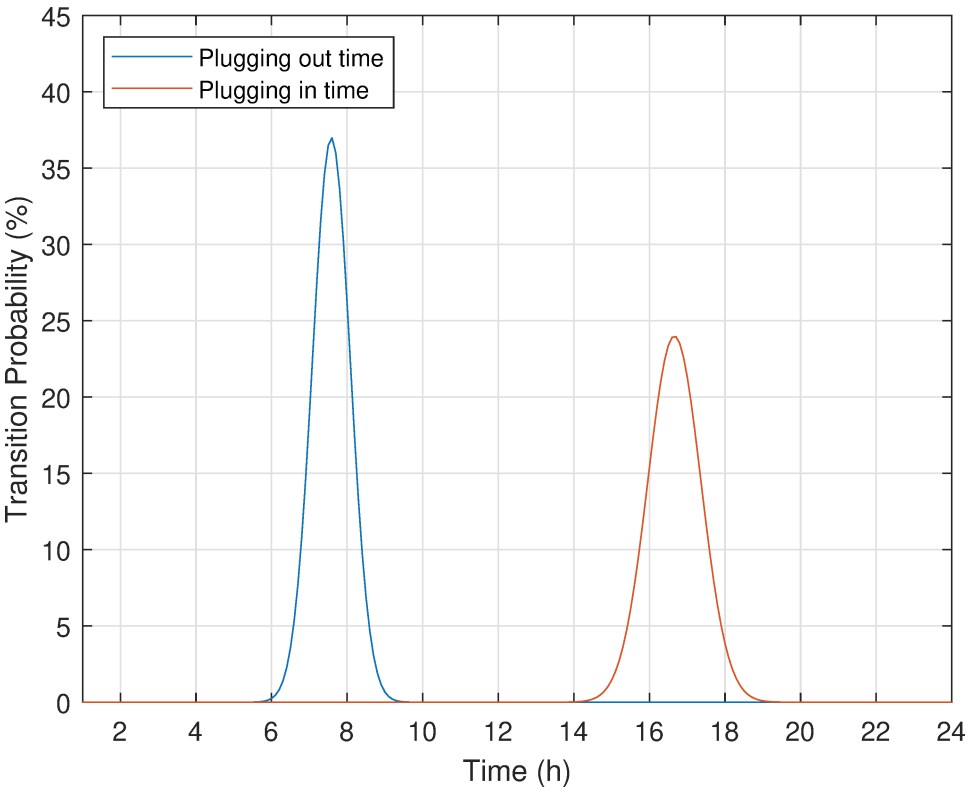

**Fig 4. Probability of PHEV's time of departure and time of arrival.**

## 3 Mathematical modeling

The proposed structure for SREH, which is optimized by smart HEMS (SHEMS), is presented in Fig. 5. The REH includes various components, such as CHP, which converts gas energy into heat and electrical energy; RES, which produces electrical energy; electric chillers and absorption chillers that fulfill cooling demand; and a boiler that generates heat using natural gas. To better fulfill REH's load demand, electrical loads are further divided into two types: shiftable and non-shiftable. Shiftable loads are flexible and can operate at any time of the day. These include appliances such as dishwashers, washing machines, and water pumps. On the other hand, one must operate non-shiftable loads as needed, without the flexibility to shift them to a different time of day. These include household appliances such as lighting and heating/cooling systems.

### 3.1 Objective function

The total operational cost of SREH is calculated using the objective function given in (5).

$$\text{Cost} = \sum_{t=1}^{24} \sum_{s=1}^{10} \psi_s \left[ \lambda_e(t,s) P_{\text{Grid}}^E(t,s) + \lambda_g(t,s) P_{\text{in}}^{\text{GAS}}(t,s) + \lambda_w P_{\text{wind}}^e(t,s) + \lambda_S P_{\text{PV-final}}^e(t,s) \right] \quad (5)$$

where $\psi_s$ is the probability of scenario $s$, $\lambda_e$ is the cost of electricity purchased from the grid, $P_{\text{Grid}}^E$ is the electrical power purchased from the grid, $\lambda_g$ is the cost of gas purchased from the grid, $P_{\text{in}}^{\text{GAS}}$ is the gas purchased from the grid, $\lambda_w$ is the operational cost of WT, $P_{\text{wind}}^e$ is the

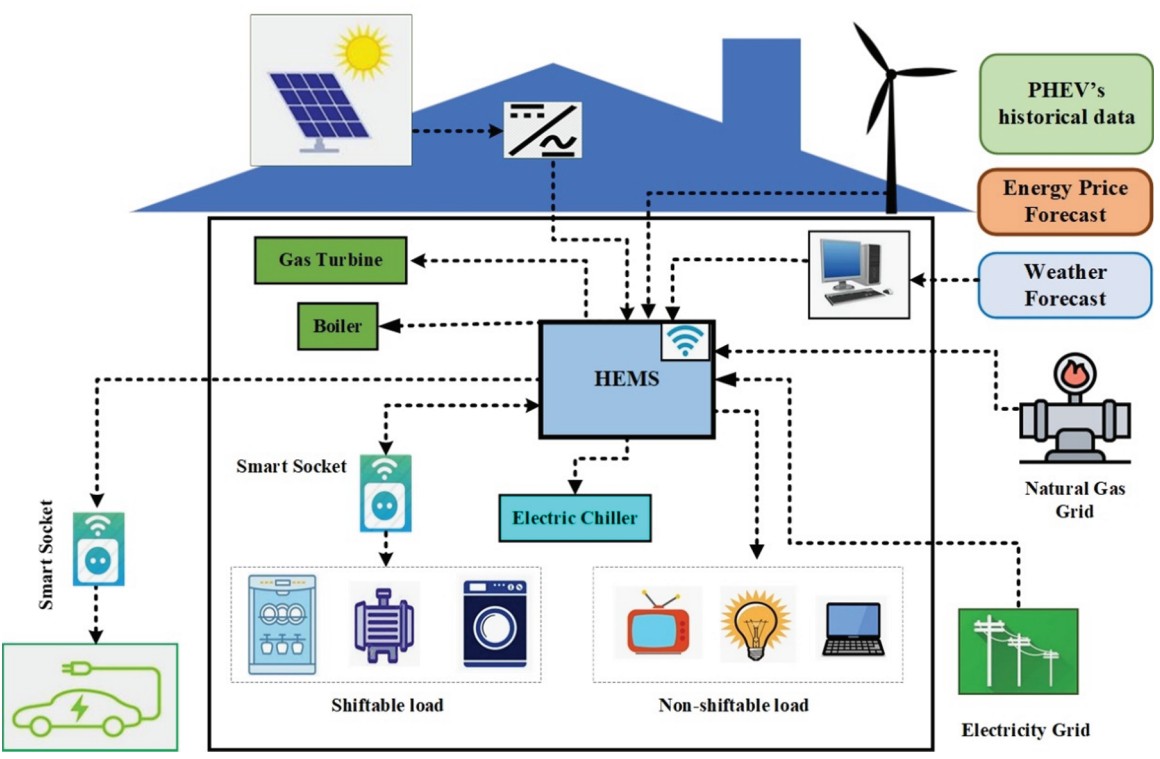

**Fig 5. Proposed model of energy management for SREH.**

electrical power produced from the WT, $\lambda_S$ is the operational cost of PV panels, and $P^e_{\text{PV-final}}$ is the electrical power produced from the PV panels.

## 3.2 Electrical components

The electrical portion of SREH received electrical power from the grid and, through a transformer, converted high-potential voltage to low-potential voltage to feed it to the loads and other components of SREH. The equation for the conversion of high voltage to low voltage is given by (6).

$$P^E_{\text{in}}(t,s) = \eta^E_{\text{Trans}} \cdot P^E_{\text{Grid}}(t,s) \tag{6}$$

where $P^E_{\text{in}}$ is the electrical power output of the transformer, $\eta^E_{\text{Trans}}$ is the efficiency of the transformer, and $P^E_{\text{Grid}}$ is the purchased electrical power from the grid.

Using an inverter, the solar power generated from PVs is converted from DC to AC to feed it to the electrical portion of SREH and is denoted by (7).

$$P^e_{\text{pv-final}}(t,s) = \eta^{\text{pv}}_{\text{DC/AC}} \cdot P^e_{\text{pv}}(t,s) \tag{7}$$

where $P^e_{\text{pv-final}}$ is the final production of PV panels in AC, $\eta^{\text{pv}}_{\text{DC/AC}}$ is the efficiency of the inverter to convert DC to AC, and $P^e_{\text{pv}}$ is the production of DC electrical power of PV panels.

The conversion of gas to electrical power using a combined heat plant (CHP) unit is denoted by (8).

$$P^e_{\text{CHP}}(t,s) = \eta^e_{\text{CHP}} \cdot P^{\text{gas}}_{\text{CHP}}(t,s) \tag{8}$$

where $P^e_{\text{CHP}}$ is the electrical power produced by CHP, $\eta^e_{\text{CHP}}$ is the efficiency of CHP to produce electrical power from gas, and $P^{\text{gas}}_{\text{CHP}}$ is the power consumed by CHP.

The minimum and the maximum limits of buying electrical power from the grid are given by (9).

$$0 \leq P^E_{\text{Grid}}(t,s) \leq P^E_{\text{Gridmax}} \tag{9}$$

where $P^E_{\text{Gridmax}}$ is the maximum power that can be purchased from the grid at any interval of time.

The permissible limits for CHP and electric chillers are given by Eqs. (10) and (11) respectively.

$$0 \leq P^e_{\text{CHP}}(t,s) \leq P^e_{\text{CHP max}} \tag{10}$$

$$0 \leq P^e_{\text{chill}}(t,s) \leq P^e_{\text{chill max}} \tag{11}$$

where $P^e_{\text{CHP max}}$ and $P^e_{\text{chill max}}$ are the maximum electrical powers that can be produced from CHP and electric chillers, respectively.

## 3.3 Heating components

A boiler and CHP use the gas input to the SREH to generate electrical and heat power. It is denoted by (12).

$$P^{\text{GAS}}_{\text{in}}(t,s) = P^{\text{gas}}_{\text{CHP}}(t,s) + P^{\text{gas}}_{\text{Boil}}(t,s) \tag{12}$$

where $P^{\text{gas}}_{\text{CHP}}$ is the gas power consumed by CHP to produce heat and electrical power, and $P^{\text{gas}}_{\text{Boil}}$ is the gas consumed by the boiler to produce heating power.

The conversion of gas to heating power through CHP is denoted by (13).

$$P^H_{\text{CHP}_h}(t,s) = \eta^H_{\text{CHP}_h} \cdot P^{\text{gas}}_{\text{CHP}}(t,s) \tag{13}$$

where $P^H_{\text{CHP}_h}$ is the heating power generated by CHP, and $\eta^H_{\text{CHP}_h}$ is the heat generation efficiency of CHP.

The gas is also used by the boiler to produce heating power. The conversion of gas to heating power through the boiler is given in (14).

$$P^H_{\text{Boil}}(t,s) = \eta^H_{\text{Boil}} \cdot P^{\text{gas}}_{\text{Boil}}(t,s) \tag{14}$$

where $P^H_{\text{Boil}}$ denotes the heating power by boiler, and $\eta^H_{\text{Boil}}$ is the heat generation efficiency of the boiler.

The minimum and the maximum limits of buying gas from grid are given by (15).

$$0 \leq P^{\text{GAS}}_{\text{in}}(t,s) \leq P^{\text{GAS}}_{\text{in max}} \tag{15}$$

where $P^{\text{GAS}}_{\text{in max}}$ is the maximum gas bought from gas grid in an interval of time.

The allowable limit for the boiler is given by (16).

$$0 \leq P^H_{\text{Boil}}(t,s) \leq P^H_{\text{Boil max}} \tag{16}$$

where $P^H_{\text{Boil max}}$ is the maximum limit of the boiler to produce heating power in any interval of time.

### 3.4 Cooling components

To meet some of the cooling load, an electric chiller converts electric power into cooling power. The conversion of electrical power to cooling power through an electric chiller is given by (17).

$$P_{\text{ec}}^{C}(t,s) = \eta_{\text{chill}}^{C} \cdot P_{\text{chill}}^{e}(t,s) \tag{17}$$

where $P_{\text{ec}}^{C}$ is the cooling power by the electric chiller, $\eta_{\text{chill}}^{C}$ is the electric chiller efficiency to convert electric power into cooling power, and $P_{\text{chill}}^{e}$ represents an electrical power used by the chiller for a cooling purpose.

The absorption chiller converts the heating power into cooling power. The conversion of heating power into cooling power through an absorption chiller is given by (18).

$$P_{\text{ac}}^{C}(t,s) = \eta_{\text{ac}}^{C} \cdot P_{\text{ac}}^{H}(t,s) \tag{18}$$

where $P_{\text{ac}}^{C}$ is the cooling power produced by the absorption chiller, $\eta_{\text{ac}}^{C}$ is the absorption chiller efficiency to convert electric power into cooling power, and $P_{\text{ac}}^{H}$ leads to the heating power used by absorption chiller.

The allowable limit for the absorption chiller is given by (19).

$$0 \le P_{\text{ac}}^{C}(t,s) \le P_{\text{ac max}}^{C} \tag{19}$$

where $P_{\text{ac max}}^{C}$ is the maximum cooling power produced by the absorption chiller at an interval of time.

### 3.5 Load balancing constraints

The electric, heating, and cooling requirements of the SREH must be satisfied at each time interval. The constraints for these loads are specified by Eqs. (20) to (22), respectively.

$$\text{Load}^{E}(t,s) + P_{\text{chill}}^{e}(t,s) + P_{\text{Ch}}^{\text{EV}}(t,s) + P_{\text{high.DR}}^{E}(t,s) = P_{\text{in}}^{E}(t,s) + P_{\text{CHP}}^{e}(t,s) +$$
$$P_{\text{wind}}^{e}(t,s) + P_{\text{pv-final}}^{e}(t,s) + P_{\text{low.DR}}^{E}(t,s) + P_{\text{Dis}}^{\text{EV}}(t,s) \tag{20}$$

where $\text{Load}^{E}$ is the electrical load, $P_{\text{chill}}^{e}$ is the electrical power consumed by the electric chiller, $P_{\text{Ch}}^{\text{EV}}$ is the electrical power consumed by the PHEV in charging its battery, $P_{\text{high.DR}}^{E}$ is the electrical load shifted by DSM to peak hours, $P_{\text{in}}^{E}$ is the electrical power purchased from the grid, $P_{\text{CHP}}^{e}$ is the electrical power produced by the CHP, $P_{\text{wind}}^{e}$ is the electrical power produced by the WT, $P_{\text{pv-final}}^{e}$ is the final electrical power produced by the PV panels, $P_{\text{low.DR}}^{E}$ is the electrical load shifted by DSM to off-peak hours, and $P_{\text{Dis}}^{\text{EV}}$ is the electrical power available from the PHEV battery during its discharge.

$$\text{Load}^{H}(t,s) + P_{\text{ac}}^{H}(t,s) + P_{\text{high.DR}}^{H}(t,s) = P_{\text{CHP}}^{H}(t,s) + P_{\text{Boil}}^{H}(t,s) + P_{\text{low.DR}}^{H}(t,s) \tag{21}$$

where $\text{Load}^{H}$ is the heating load, $P_{\text{ac}}^{H}$ is the heating power consumed by the absorption chiller, $P_{\text{high.DR}}^{H}$ is the shifted heating power by DSM to peak hours, $P_{\text{CHP}}^{H}$ is the heating power produced by the CHP, $P_{\text{Boil}}^{H}$ is the heating power produced by the boiler, and $P_{\text{low.DR}}^{H}$ is the shifted heating power by DSM to off-peak hours.

$$\text{Load}^{C}(t,s) + P_{\text{high.DR}}^{C}(t,s) = P_{\text{ac}}^{C}(t,s) + P_{\text{ec}}^{C}(t,s) + P_{\text{low.DR}}^{C}(t,s) \tag{22}$$

where $\text{Load}^C$ is the cooling load, $P^C_{\text{high.DR}}$ is the shifted cooling power by DSM to peak hours, $P^C_{\text{ac}}$ is the cooling power produced by the absorption chiller, $P^C_{\text{ec}}$ is the cooling power produced by the electric chiller, and $P^C_{\text{low.DR}}$ is the shifted cooling power by DSM to off-peak hours.

## 3.6 DSM constraints

DSM enables the shifting of controllable loads from peak to off-peak hours to lower the operational costs of the REH without compromising consumer comfort [82]. Controllable devices in the SREH allow for required shifts without compromising performance. This work computes the operational cost over a 24-hour period, and we must add the load shed from one hour to the next to maintain the same load usage over the 24-hour horizon. This paper applies DSM to electric, heating, and cooling loads. The DSM on electrical load is given by Eqs. (23) to (26).

$$\sum_{t=1}^{24} P^E_{\text{high.DR}}(t,s) = \sum_{t=1}^{24} P^E_{\text{low.DR}}(t,s) \tag{23}$$

$$0 \le P^E_{\text{high.DR}}(t,s) \le P^E_{\text{DR-high}} \cdot \text{Load}^E(t,s) \cdot I^E_{\text{high}}(t,s) \tag{24}$$

$$0 \le P^E_{\text{low.DR}}(t,s) \le P^E_{\text{DR-low}} \cdot \text{Load}^E(t,s) \cdot I^E_{\text{low}}(t,s) \tag{25}$$

$$0 \le I^E_{\text{high}}(t,s) + I^E_{\text{low}}(t,s) \le 1 \tag{26}$$

where $P^E_{\text{high.DR}}$ is the electrical load shifted to peak hours, $P^E_{\text{low.DR}}$ is the electrical load shifted to off-peak hours, $P^E_{\text{DR-high}}$ and $P^E_{\text{DR-low}}$ are scaling factors for the electrical load that can be shifted to peak and off-peak hours, respectively. $I^E_{\text{high}}$ and $I^E_{\text{low}}$ are binary variables determining if a load is shifted to peak hour or off-peak hour, respectively.

The DSM on heating load is given by Eqs. (27) to (30).

$$\sum_{t=1}^{24} P^H_{\text{high.DR}}(t,s) = \sum_{t=1}^{24} P^H_{\text{low.DR}}(t,s) \tag{27}$$

$$0 \le P^H_{\text{high.DR}}(t,s) \le P^H_{\text{DR-high}} \cdot \text{Load}^H(t,s) \cdot I^H_{\text{high}}(t,s) \tag{28}$$

$$0 \le P^H_{\text{low.DR}}(t,s) \le P^H_{\text{DR-low}} \cdot \text{Load}^H(t,s) \cdot I^H_{\text{low}}(t,s) \tag{29}$$

$$0 \le I^H_{\text{high}}(t,s) + I^H_{\text{low}}(t,s) \le 1 \tag{30}$$

where $P^H_{\text{high.DR}}$ is the heating load shifted to peak hours, $P^H_{\text{low.DR}}$ is the heating load shifted to off-peak hours, $P^H_{\text{DR-high}}$ and $P^H_{\text{DR-low}}$ are scaling factors for the heating load that can be shifted to peak and off-peak hours, respectively. $I^H_{\text{high}}$ and $I^H_{\text{low}}$ are binary variables determining if a load is shifted to peak hour or off-peak hour, respectively.

The DSM on cooling load is given by Eqs. (31) to (34).

$$\sum_{t=1}^{24} P^C_{\text{high.DR}}(t,s) = \sum_{t=1}^{24} P^C_{\text{low.DR}}(t,s) \tag{31}$$

$$0 \le P^C_{\text{high.DR}}(t,s) \le P^C_{\text{DR-high}} \cdot \text{Load}^C(t,s) \cdot I^C_{\text{high}}(t,s) \tag{32}$$

$$0 \leq P_{\text{low.DR}}^{C}(t,s) \leq P_{\text{DR-low}}^{C} \cdot \text{Load}^{C}(t,s) \cdot I_{\text{low}}^{C}(t,s) \tag{33}$$

$$0 \leq I_{\text{high}}^{C}(t,s) + I_{\text{low}}^{C}(t,s) \leq 1 \tag{34}$$

where $P_{\text{high.DR}}^{C}$ is the cooling load shifted to peak hours, $P_{\text{low.DR}}^{C}$ is the cooling load shifted to off-peak hours, $P_{\text{DR-high}}^{C}$ and $P_{\text{DR-low}}^{C}$ are scaling factors for the cooling load that can be shifted to peak and off-peak hours, respectively. $I_{\text{high}}^{C}$ and $I_{\text{low}}^{C}$ are binary variables determining if a load is shifted to peak hour or off-peak hour, respectively.

## 3.7 PHEV modeling

In this study, a DC/AC converter connects the PHEV's battery to the SREH, enabling either one-way or two-way electrical power exchange, depending on the situation. The PHEV is available at home for some period of time depending upon the arrival and departure times of the consumer. The level of charge of PHEV after the arrival time depends upon the traveling distance of the consumer. The power usage of the PHEV depends on the driving distance and efficiency of the vehicle and is given by (35).

$$P_{\text{consp}}^{\text{PHEV}} = \eta_v \cdot DD \tag{35}$$

where $P_{\text{consp}}^{\text{PHEV}}$ is the electrical power consumed by the PHEV in traveling, $\eta_v$ is the mileage efficiency of the PHEV, and $DD$ is the driving distance traveled by the PHEV.

The charging or discharging rate of PHEV must not exceed the maximum charge or discharge allowable limit of the PHEV battery and is given by Eqs. (36) and (37), respectively.

$$P_{\text{Ch}}^{\text{PHEV}}(t,s) \leq P_{\text{max}}^{\text{Ch}} \tag{36}$$

$$P_{\text{Dis}}^{\text{PHEV}}(t,s) \leq P_{\text{max}}^{\text{Dis}} \tag{37}$$

where $P_{\text{Ch}}^{\text{PHEV}}$ and $P_{\text{Dis}}^{\text{PHEV}}$ are the charging and discharging power of the PHEV battery, and $P_{\text{max}}^{\text{Ch}}$ and $P_{\text{max}}^{\text{Dis}}$ are the maximum limits of charging and discharging of the PHEV battery, respectively.

The electrical energy stored in a PHEV's battery must not exceed the battery's allowable limit and is given by (38).

$$0 \leq P_{\text{elec}}^{\text{PHEV}}(t,s) \leq \text{Cap}^{\text{PHEV}} \tag{38}$$

where $P_{\text{elec}}^{\text{PHEV}}$ is the electrical power stored in the PHEV battery at any interval of time, and $\text{Cap}^{\text{PHEV}}$ is the maximum capacity of the PHEV battery.

For the comfort of the consumer, the charge level of the battery of a PHEV must be at least 50% of the maximum capacity when the consumer departs, which is given by (39).

$$P_{\text{elec}}^{\text{PHEV}}(DT(t,s) - 1) = 0.5 \cdot \text{Cap}^{\text{PHEV}} \tag{39}$$

The schedule of charge of PHEV is calculated by (40).

$$\text{SOC}^{\text{PHEV}}(t,s) = \frac{P_{\text{elec}}^{\text{PHEV}}(t,s)}{\text{Cap}^{\text{PHEV}}} \tag{40}$$

where SOC$^{\text{PHEV}}$ is the schedule of charge of the PHEV battery at any interval of time.

The SOC of the PHEV must be in the allowable range according to its technical specifications. This is given in (41).

$$\text{SOC}_{\min}^{\text{PHEV}} \leq \text{SOC}^{\text{PHEV}}(t,s) \leq \text{SOC}_{\max}^{\text{PHEV}} \tag{41}$$

where SOC$_{\min}^{\text{PHEV}}$ and SOC$_{\max}^{\text{PHEV}}$ are the minimum and maximum schedule of charge of the PHEV battery, respectively.

The difference between a regular battery and a PHEV battery is that a regular battery is available all day, making it easier to model. In contrast, a PHEV battery is only available when the vehicle is at the building. Additionally, a PHEV battery discharges during travel, whereas a conventional battery only discharges to meet the demands of the REH during peak hours. There are also additional constraints associated with a PHEV battery, such as the need for it to be fully charged before departure time to ensure the comfort level of the driver. These additional constraints need to be managed effectively by an intelligent charging mechanism.

## 4 Methodology

Energy systems frequently employ MILP modeling for complicated systems. A three-step iterative technique is commonly employed to design MILPs [83]. The initial stage entails delineating a collection of decision factors that signify the options requiring optimization within the framework. Often, the second phase involves defining the model's restrictions, and the third step requires defining the goal function. However, it can execute these two processes in any order [84].

During the modeling building process, it frequently becomes apparent that the initially specified collection of variables for decision-making is insufficient. It is often necessary to explicitly describe choice factors that seem like implicit outcomes of other choices. The incorporation of additional factors following a failed endeavor to establish limits and objectives constitutes the "loop" in the method [85]. The accurate description of decision factors may prove particularly complex when modeling using integer parameters. By using binary values in a framework, we can represent binary choices, enforce conditional assertions, and incorporate certain nonlinearities into the system, all of which we can convert into a comparable MILP.

The GAMS is a sophisticated computing framework for scientific optimization. The purpose of GAMS is to simulate and resolve optimization challenges that are linear, unpredictable, and mixed-integer. Users can construct extensive, stable simulations, adaptable to new circumstances, thanks to the system's design for intricate and massive modeling tasks. The system is accessible on several computer systems. Models are transferable across many platforms [86]. The concept of a logical methodology for representing, manipulating, and solving extensive mathematical problems integrates traditional and contemporary concepts into a coherent and analytically feasible framework. The use of generator arrays in linear programming highlighted the importance of consistently labeling rows and columns. The link to the nascent relational data paradigm became apparent. Knowledge of conventional coding languages for managing name-spaces naturally fosters a mindset oriented toward sets and tuples, which subsequently results in the database system [87].

The proposed method for the optimization of SREH in the presence of PHEV with DSM is shown in Fig. 6. The optimization of REH starts with the collection of data like electricity price, gas price, electric demand, heating demand, and cooling demand. For stochastic modeling of solar and wind energy, historical data on solar irradiance and wind speed is essential.

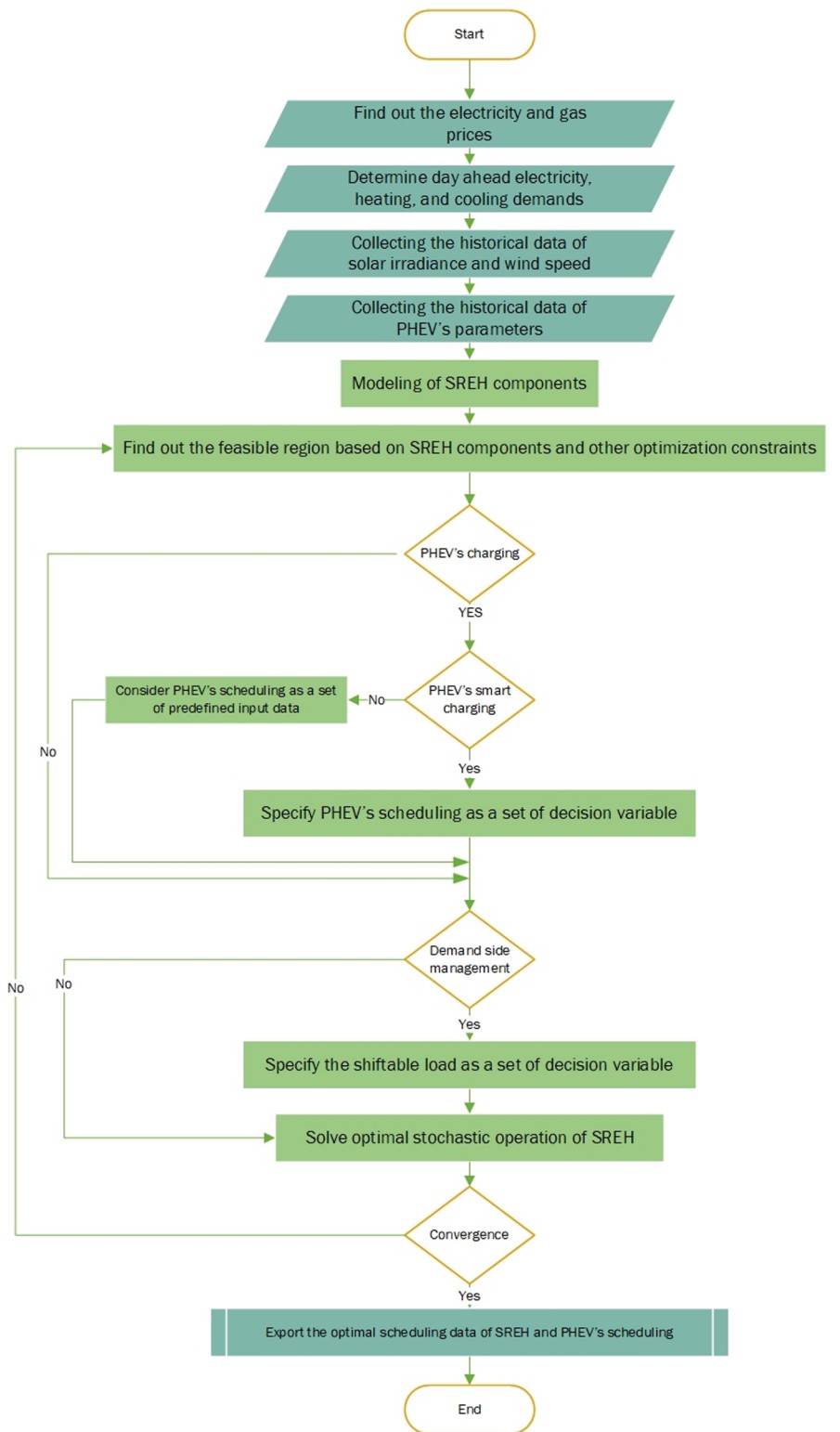

**Fig 6. Flow chart of the proposed method for optimal operation of SREH in the presence of PHEV.**

Additionally, accurate modeling of PHEV availability and charging requires historical data on parameters such as departure time, arrival time, and traveling distance. Once all the relevant data has been gathered, the different parts needed to change one type of energy into another in REH are modeled as a MILP and solved with GAMS software. Then, based upon the constraints of REH components, like maximum and minimum input or output, its feasible operating region is identified. Next, we incorporate PHEV's charging, PHEV's smart charging, and DSM modules to enhance REH's intelligence. Finally, this work solved the entire SREH model, export the results, and conduct further investigations.

## 5 Simulation and case studies

To check the impact of the integration of PHEVs into the REH and the significance of smart charging in the SREH environment, four different cases are discussed in this section.

1. **Case Study-I:** The first case study is the foundational scenario excluding the participation of PHEV in REH.
2. **Case Study-II:** The second case study involves PHEV in REH.
3. **Case Study-III:** The third scenario pertains to the intelligent charging of PHEVs in smart residential energy network.
4. **Case Study-IV:** The fourth case study examines the smart PHEVs within the smart residential energy network in conjunction with DSM.

### 5.1 Simulation setup

The proposed SREH model for scenario-based optimal scheduling to a residential building that includes two RESs (solar and wind), a CCHP unit, an electric chiller, a boiler, a PHEV, and various electrical, heating, and cooling loads. This study considers 10% of the load to be shiftable for DSM. The electrical, heating, and cooling demands considered for SREH are illustrated in Fig. 7.

In this work, the PHEV considered for simulation purposes is the Chevrolet Volt, which has a 33000 Wh electric battery. The detailed technical features of this PHEV are given in Table 2.

For the REH simulation, Table 3 shows the features of the EH central devices, Table 4 shows the features of the PV modules, and Table 5 shows the features of the wind turbines [76].

The electricity and natural gas price used for simulation is taken from [88] and are shown in Fig. 8.

## 6 Results

The present work formulate the overall problem as MILP and solve it in the GAMS environment, incorporating RES. It simulate four different scenarios to evaluate the SREH's performance in the context of PHEV.

### 6.1 Case Study-I

In case study I, this work investigate the optimal operation of REH, which includes RES (both solar and wind energy), CHP, EC, AC, a boiler, and three types of loads. The electricity and gas procured from the grid to fulfill demand are shown in Table 6 (Case Study-I(a)). Given the lower price of gas compared to electricity, we purchase more gas than electricity to ensure optimal operation of the REH. The REH requires more gas from 11:00 to 19:00, but it

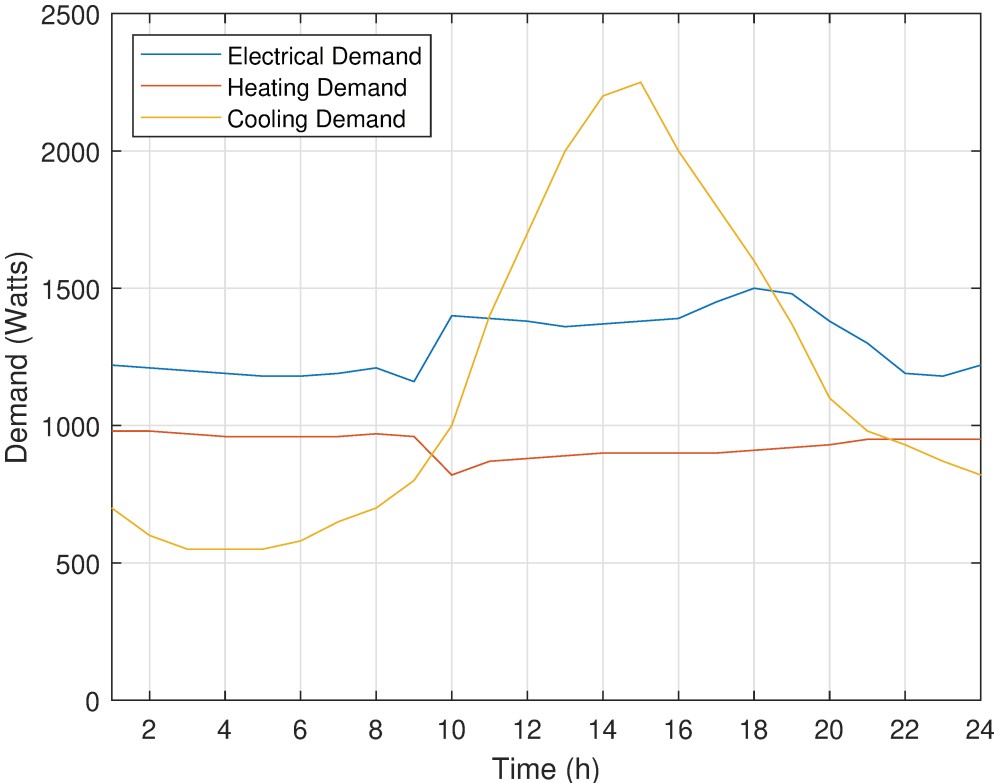

**Fig 7. Electrical, heating, and cooling demands for SREH.**

**Table 2. Technical features of PHEV battery.**

| Parameter | Unit | Value |
|---|---|---|
| Name of PHEV | Chevrolet | Volt |
| Battery Capacity | kWh | 33 |
| Max. charging/discharging rate | kW | 3.3 |
| Plug out time | h | 07 |
| Plug in time | h | 19 |
| Consumption per km | kWh | 0.1 |
| Battery lower limit | % | 20 |
| Battery upper limit | % | 100 |

restricts its procurement to 2.4 kW to prevent exceeding the upper limit. During this period, the dependency on the electricity increases. The total operating cost of REH in Case Study-I is 7675.00 USD for a 24-hour period.

The electrical demand of REH and electrical energy consumed by electric chiller and EHP is fulfilled by the electrical energy purchased from the grid, the electrical energy produced from RES (solar and wind), and the electrical energy generated by the CHP, as shown in Table 6 (Case Study-I(b)). The CHP utilizes the lower cost of gas compared to electricity, operates at its peak efficiency, and generates 480 watts of electrical power from the gas. The electrical power generated from RES is non-dispatchable and available only when sunlight or wind is available.

**Table 3. Technical features of central devices.**

| Parameter | Unit | Value |
|---|---|---|
| Efficiency of gas turbines in heat generation | % | 40 |
| Efficiency of gas turbines in electricity generation | % | 35 |
| Boiler efficiency | % | 80 |
| Absorption chiller COP | % | 130 |
| Electric Chiller efficiency | % | 80 |
| Maximum heat power generated from boiler | kW | 1.8 |
| Transformer efficiency | % | 95 |
| Maximum power generated from CHP | kW | 1.2 |

**Table 4. Technical features of PV module.**

| Parameter | Unit | Value |
|---|---|---|
| Maximum PV module output power | kW | 0.4 |
| Derating factor of PV module | % | 80 |
| Solar irradiance at standard test conditions | kW/m$^2$ | 1 |
| Coefficient of temperature | %/°C | -0.5 |
| Temperature of PV cell | °C | 60 |
| Temperature of PV cell under STC | °C | 25 |

**Table 5. Technical features of wind turbine.**

| Parameter | Unit | Value |
|---|---|---|
| Maximum WT output | kW | 0.4 |
| Cut in speed of WT | m/s | 4 |
| Cut out speed of WT | m/s | 22 |
| Rated speed of WT | m/s | 10 |

The heating demand of REH is met by gas turbine and boiler only, as shown in Table 6 (Case Study-I(c)). Depending upon the weather and load condition, EHP can produce heating power by changing its mode of operation, but in this case, it produces only cooling energy because cooling demand is significantly higher than the heating demand. EHP cannot produce heating and cooling power simultaneously.

The cooling demand of the REH is fulfilled by an absorption chiller, an EHP, and an electric chiller, as illustrated in Table 6 (Case Study-I(d)). The EHP generates cooling power continuously due to its low cost, while electric chillers only activate during peak hours to meet the cooling demand, as they require expensive electricity.

## 6.2 Case Study-II

This case study incorporates the PHEV into the REH, complementing the components already present in case 1. We assume that PHEVs depart from their residence at 7:00 and reach their destination at 17:00, using half of their battery capacity during the journey. In this scenario, the unmanaged charging of PHEV leads to an increase in electrical demand during peak hours. This situation results in an increase in the operating cost of REH compared to the first case, where PHEV was not present. To meet the additional electrical demand due to PHEV charging, the amount of purchased electricity and gas increases, as shown in Table 7 (Case Study-II(a)).

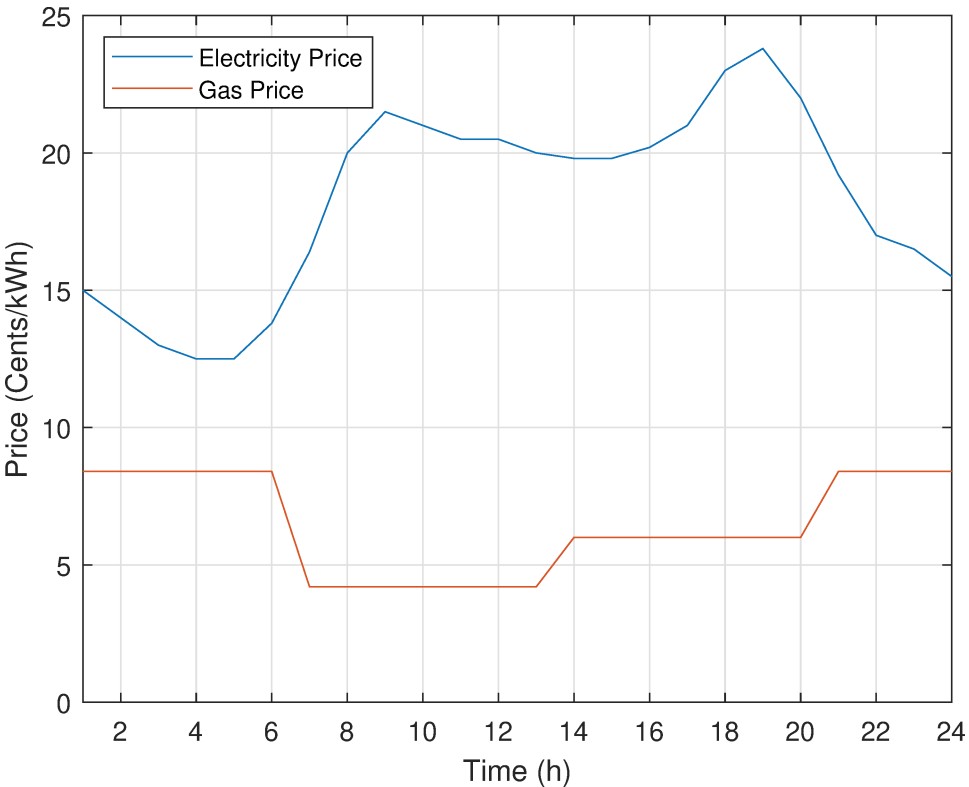

**Fig 8. Electricity and gas prices.**

The electric, heating, and cooling portions of REH in Case Study-II are shown in Table 7 (Case Study-II(b,c, and d)), respectively. The unmanaged charging of PHEVs creates an increased burden on REH, mainly in peak hours. The REH's operational cost rises to 7723.70 USD due to the ineffective use of PHEV's stored electrical energy during peak hours. Therefore, adding a PHEV to the REH without any management increases the operational cost by 0.63%.

The SOC and charging of PHEV without any management are shown in Fig. 9. The PHEV gets charged immediately after coming home, irrespective of the fact that it is peak hour time. The unmanaged charging of PHEV has caused a significant increase in the energy costs due to higher dependency on costly electrical energy and gas in peak hours. In addition, REH experiences increased electrical peak demands because charging of the PHEV is not coordinated. The ability of PHEV to act as a storage resource is untapped in this case, thus reducing the operational efficiency of REH. Thus, adding a PHEV to the REH without any smart management system has increased operational cost, increased peak demands, and decreased reliability. To improve this situation, implementing a smart charging strategy could significantly reduce the operational cost of REH and also improve the flow of energy within REH, which is discussed in the next case study.

### 6.3 Case Study-III

In Case Study-III, a smart management system is implemented for the charging of PHEV, which not only manages peak/off peak hours but also provides electrical energy to the REH

**Table 6. Results for Case Study-I.**

**Case Study-I (a)**

| Hour | 1 | 2 | 3 | 4 | 5 | 6 | 7 | 8 | 9 | 10 | 11 | 12 |
|---|---|---|---|---|---|---|---|---|---|---|---|---|
| Electricity Purchased (Watts) | 652.6 | 600 | 568.4 | 557.9 | 369.4 | 318.7 | 451.2 | 646 | 719 | 867.5 | 968.7 | 1182.5 |
| Gas Purchased (Watts) | 1882.2 | 1882.2 | 1811.1 | 1800 | 1800 | 1800 | 1800 | 1811.1 | 1800 | 1644.4 | 2400 | 2400 |
| Hour | 13 | 14 | 15 | 16 | 17 | 18 | 19 | 20 | 21 | 22 | 23 | 24 |
| Electricity Purchased (Watts) | 1591.9 | 1888 | 1991.1 | 1708.2 | 1503.7 | 1403.6 | 1234.7 | 1157.8 | 910.8 | 717.8 | 682.1 | 703.1 |
| Gas Purchased (Watts) | 2400 | 2400 | 2400 | 2400 | 2400 | 2400 | 2400 | 2084.1 | 1788.8 | 1788.8 | 1788.8 | 1788.8 |

**Case Study-I (b)**

| Hour | 1 | 2 | 3 | 4 | 5 | 6 | 7 | 8 | 9 | 10 | 11 | 12 |
|---|---|---|---|---|---|---|---|---|---|---|---|---|
| Electricity from CHP (Watt) | 480 | 480 | 480 | 480 | 480 | 480 | 480 | 480 | 480 | 480 | 480 | 480 |
| Electricity from grid (Watt) | 589 | 541.5 | 513 | 503.4 | 333.4 | 287.7 | 407.2 | 583 | 648.8 | 782.9 | 874.3 | 1067.2 |
| RES power production (Watt) | 400 | 400 | 400 | 400 | 569 | 629.1 | 541.3 | 396.2 | 316.9 | 495.8 | 614 | 780.3 |
| Electric chiller consumption (Watt) | 0 | 0 | 0 | 0 | 0 | 0 | 0 | 0 | 0 | 179.5 | 483 | 786.5 |
| Heat pump consumption (Watt) | 280 | 240 | 220 | 220 | 220 | 232 | 260 | 280 | 320 | 400 | 400 | 400 |
| Hour | 13 | 14 | 15 | 16 | 17 | 18 | 19 | 20 | 21 | 22 | 23 | 24 |
| Electricity from CHP (Watt) | 480 | 480 | 480 | 480 | 480 | 480 | 480 | 480 | 480 | 480 | 480 | 480 |
| Electricity from grid (Watt) | 1436.7 | 1703.9 | 1796.9 | 1541.6 | 1357.1 | 1266.7 | 1114.3 | 1045 | 822 | 647.8 | 615.6 | 634.6 |
| RES power production (Watt) | 750.7 | 733.8 | 708.5 | 674.6 | 678.9 | 578.4 | 435.7 | 200 | 346.7 | 400 | 400 | 400 |
| Electric chiller consumption (Watt) | 990 | 1040 | 790 | 590 | 393.5 | 167 | 0 | 0 | 0 | 0 | 0 | 0 |
| Heat pump consumption (Watt) | 400 | 400 | 400 | 400 | 400 | 400 | 400 | 400 | 392 | 372 | 348 | 328 |

**Case Study-I (c)**

| Hour | 1 | 2 | 3 | 4 | 5 | 6 | 7 | 8 | 9 | 10 | 11 | 12 |
|---|---|---|---|---|---|---|---|---|---|---|---|---|
| Gas turbine heating power (Watt) | 420 | 420 | 420 | 420 | 420 | 420 | 420 | 420 | 420 | 420 | 420 | 420 |
| Boiler heating power (Watt) | 560 | 560 | 550 | 540 | 540 | 540 | 540 | 550 | 540 | 400 | 450 | 460 |
| Heat pump heating power (Watt) | 0 | 0 | 0 | 0 | 0 | 0 | 0 | 0 | 0 | 0 | 0 | 0 |
| Hour | 13 | 14 | 15 | 16 | 17 | 18 | 19 | 20 | 21 | 22 | 23 | 24 |
| Gas turbine heating power (Watt) | 420 | 420 | 420 | 420 | 420 | 420 | 420 | 420 | 420 | 420 | 420 | 420 |
| Boiler heating power (Watt) | 470 | 480 | 480 | 480 | 480 | 490 | 500 | 510 | 530 | 530 | 530 | 530 |
| Heat pump heating power (Watt) | 0 | 0 | 0 | 0 | 0 | 0 | 0 | 0 | 0 | 0 | 0 | 0 |

**Case Study-I (d)**

| Hour | 1 | 2 | 3 | 4 | 5 | 6 | 7 | 8 | 9 | 10 | 11 | 12 |
|---|---|---|---|---|---|---|---|---|---|---|---|---|
| Absorption chiller cooling power (Watt) | 0 | 0 | 0 | 0 | 0 | 0 | 0 | 0 | 0 | 0 | 220.5 | 217 |
| Heat pump cooling power (Watt) | 700 | 600 | 550 | 550 | 550 | 580 | 650 | 700 | 800 | 1000 | 1000 | 1000 |
| Electric chiller cooling power (Watt) | 0 | 0 | 0 | 0 | 0 | 0 | 0 | 0 | 0 | 0 | 179.5 | 483 |
| Hour | 13 | 14 | '15 | 16 | 17 | 18 | 19 | 20 | 21 | 22 | 23 | 24 |
| Absorption chiller cooling power (Watt) | 213.5 | 210 | 210 | 210 | 210 | 206.5 | 203 | 100 | 0 | 0 | 0 | 0 |
| Heat pump cooling power (Watt) | 1000 | 1000 | 1000 | 1000 | 1000 | 1000 | 1000 | 1000 | 980 | 930 | 870 | 820 |
| Electric chiller cooling power (Watt) | 483 | 786.5 | 990 | 1040 | 790 | 590 | 393.5 | 167 | 0 | 0 | 0 | 0 |

**Table 7. Results for Case Study-II.**

**Case Study-II (a)**

| Hour | 1 | 2 | 3 | 4 | 5 | 6 | 7 | 8 | 9 | 10 | 11 | 12 |
|---|---|---|---|---|---|---|---|---|---|---|---|---|
| Electricity Purchased (Watts) | 652.6 | 600 | 568.4 | 557.9 | 369.4 | 318.7 | 556.4 | 551.2 | 603.2 | 730.7 | 842.4 | 956.2 |
| Gas Purchased (Watts) | 1882.2 | 1882.2 | 1811.1 | 1800 | 1800 | 1800 | 1800 | 1811.1 | 1800 | 1644.4 | 2400 | 2400 |
| Hour | 13 | 14 | 15 | 16 | 17 | 18 | 19 | 20 | 21 | 22 | 23 | 24 |
| Electricity Purchased (Watts) | 1581.4 | 1898.5 | 1999.4 | 1602.9 | 1503.7 | 1403.6 | 1234.7 | 1157.8 | 216.1 | 23.15 | 0 | 273.6 |
| Gas Purchased (Watts) | 2400 | 2400 | 2400 | 2400 | 2400 | 2400 | 2400 | 2084.1 | 1788.8 | 1788.8 | 1788.8 | 1788.8 |

**Case Study-II (b)**

| Hour | 1 | 2 | 3 | 4 | 5 | 6 | 7 | 8 | 9 | 10 | 11 | 12 |
|---|---|---|---|---|---|---|---|---|---|---|---|---|
| Electricity from CHP (Watt) | 480 | 480 | 480 | 480 | 480 | 480 | 480 | 480 | 480 | 480 | 480 | 480 |
| Electricity from grid (Watt) | 620.0 | 570 | 540 | 529.9 | 350.9 | 302.8 | 528.6 | 523.7 | 573 | 694.1 | 800.3 | 908.4 |
| RES power production (Watt) | 400 | 400 | 400 | 400 | 569 | 629.1 | 541.3 | 396.2 | 316.9 | 495.8 | 614 | 780.3 |
| Electric chiller consumption (Watt) | 0 | 0 | 0 | 0 | 0 | 0 | 0 | 0 | 0 | 224.3 | 603.7 | 983.1 |
| Heat pump consumption (Watt) | 280 | 240 | 220 | 220 | 220 | 232 | 260 | 280 | 320 | 400 | 400 | 400 |
| Hour | 13 | 14 | 15 | 16 | 17 | 18 | 19 | 20 | 21 | 22 | 23 | 24 |
| Electricity from CHP (Watt) | 480 | 480 | 480 | 480 | 480 | 480 | 480 | 480 | 480 | 480 | 480 | 480 |
| Electricity from grid (Watt) | 1502.3 | 1803.6 | 1899.5 | 1522.8 | 1428.5 | 1333.4 | 1173.0 | 1100 | 205.3 | 22 | 0 | 259.9 |
| RES power production (Watt) | 750.7 | 733.8 | 708.5 | 674.6 | 678.9 | 578.4 | 435.7 | 200 | 346.7 | 400 | 400 | 400 |
| Electric chiller consumption (Watt) | 1273.5 | 1300 | 987.5 | 737.5 | 491.8 | 208.7 | 0 | 0 | 0 | 0 | 0 | 0 |
| Heat pump consumption (Watt) | 400 | 400 | 400 | 400 | 400 | 400 | 400 | 400 | 392 | 372 | 348 | 328 |

**Case Study-II (c)**

| Hour | 1 | 2 | 3 | 4 | 5 | 6 | 7 | 8 | 9 | 10 | 11 | 12 |
|---|---|---|---|---|---|---|---|---|---|---|---|---|
| Gas turbine heating power (Watt) | 360 | 360 | 360 | 360 | 360 | 360 | 4200 | 420 | 420 | 420 | 420 | 420 |
| Boiler heating power (Watt) | 620 | 620 | 610 | 600 | 600 | 600 | 540 | 550 | 540 | 400 | 40 | 460 |
| Heat pump heating power (Watt) | 0 | 0 | 0 | 0 | 0 | 0 | 0 | 0 | 0 | 0 | 0 | 0 |
| Hour | 13 | 14 | 15 | 16 | 17 | 18 | 19 | 20 | 21 | 22 | 23 | 24 |
| Gas turbine heating power (Watt) | 420 | 420 | 400 | 400 | 400 | 400 | 400 | 420 | 360 | 360 | 360 | 360 |
| Boiler heating power (Watt) | 470 | 500 | 500 | 500 | 500 | 510 | 520 | 510 | 590 | 590 | 590 | 590 |
| Heat pump heating power (Watt) | 0 | 0 | 0 | 0 | 0 | 0 | 0 | 0 | 0 | 0 | 0 | 0 |

**Case Study-II (d)**

| Hour | 1 | 2 | 3 | 4 | 5 | 6 | 7 | 8 | 9 | 10 | 11 | 12 |
|---|---|---|---|---|---|---|---|---|---|---|---|---|
| Absorption chiller cooling power (Watt) | 0 | 0 | 0 | 0 | 0 | 0 | 0 | 0 | 0 | 0 | 220.5 | 217 |
| Heat pump cooling power (Watt) | 700 | 600 | 550 | 550 | 550 | 580 | 650 | 700 | 800 | 1000 | 1000 | 1000 |
| Electric chiller cooling power (Watt) | 0 | 0 | 0 | 0 | 0 | 0 | 0 | 0 | 0 | 0 | 179.5 | 483 |
| Hour | 13 | 14 | '15 | 16 | 17 | 18 | 19 | 20 | 21 | 22 | 23 | 24 |
| Absorption chiller cooling power (Watt) | 213.5 | 210 | 210 | 210 | 210 | 206.5 | 203 | 100 | 0 | 0 | 0 | 0 |
| Heat pump cooling power (Watt) | 1000 | 1000 | 1000 | 1000 | 1000 | 1000 | 1000 | 1000 | 980 | 930 | 870 | 820 |
| Electric chiller cooling power (Watt) | 483 | 786.5 | 990 | 1040 | 790 | 590 | 393.5 | 167 | 0 | 0 | 0 | 0 |

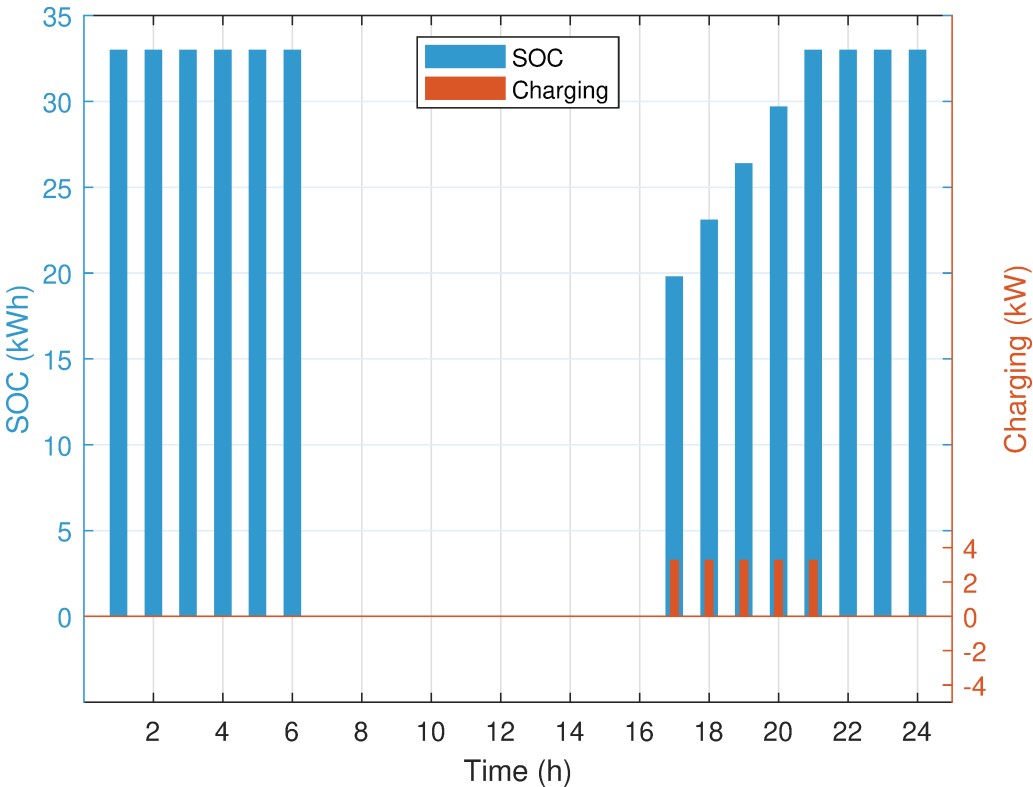

**Fig 9. SOC of PHEV battery and its charging in Case Study-II.**

by discharging when necessary. This approach optimizes the charging and discharging of PHEV battery to minimize the operational costs of REH. The smart charging/discharging system ensures that electrical energy demand is distributed more evenly throughout the day, reducing peaks and leveraging off-peak charging. It is assumed that PHEV leaves home at 7:00 with a full charge and arrives home after traveling a distance at 17:00 with 50% charge left. The electricity and gas purchased from the grids in Case Study-III are illustrated in Fig. 10.

The smart charging of PHEV makes it possible to discharge the battery of PHEV in peak hours and charge it back in off-peak hours, as shown in Fig. 11. Instead of charging immediately upon arrival, as in the previous case, the PHEV battery is discharged during peak hours to supply energy to REH during high tariffs and demand. By extracting a portion of electrical energy from the battery during peak hours and charging it during off-peak hours, we significantly reduce the operational cost of SREH. The battery is charged again during off-peak hours when electricity is cheap, ensuring that it is ready for use again the next day. Charging the PHEV during off-peak hours allows the REH to take advantage of lower energy rates. The PHEV battery functions as both an energy storage system and a power source, increasing the flexibility and efficiency of the REH. In case Study-III, smart charging reduces the operational cost of SREH to 7523.10 USD from 7723.70 USD, indicating a reduction of approximately 2.59%. The electric, heating and cooling portion of REH in Case Study III is shown in Figs 12-14. This case highlights the importance of integrating smart energy management systems for PHEVs in achieving a cost-effective and efficient REH.

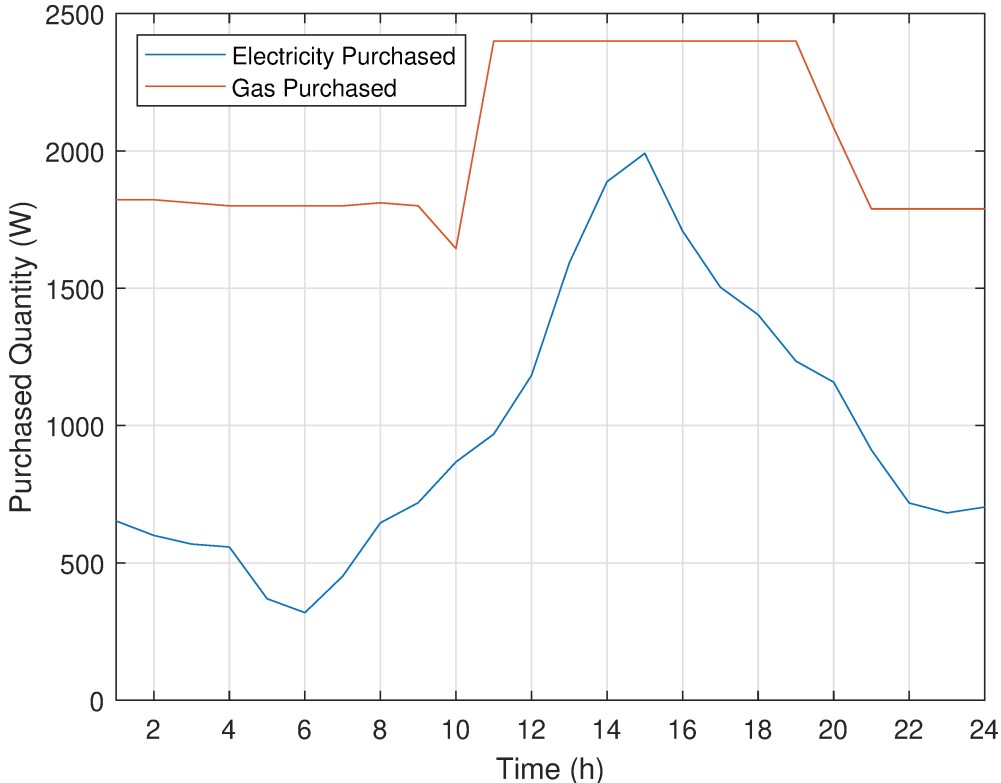

**Fig 10. Electricity and natural gas purchased from the grid in Case Study-III.**

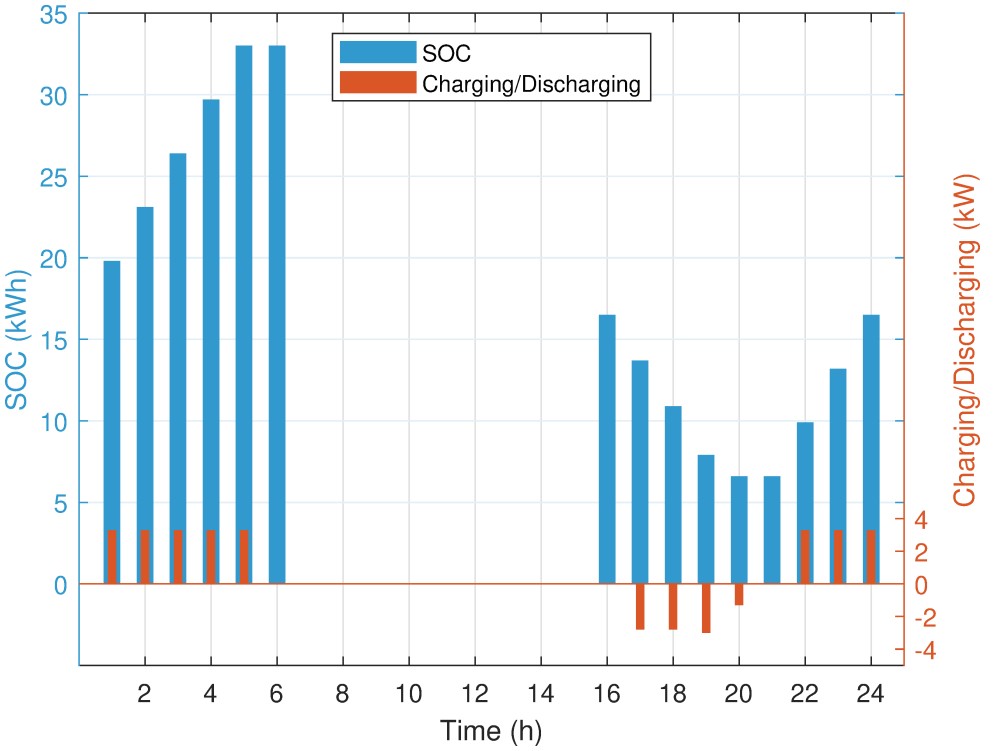

**Fig 11. SOC and charging/discharging of PHEV battery in Case Study-III.**

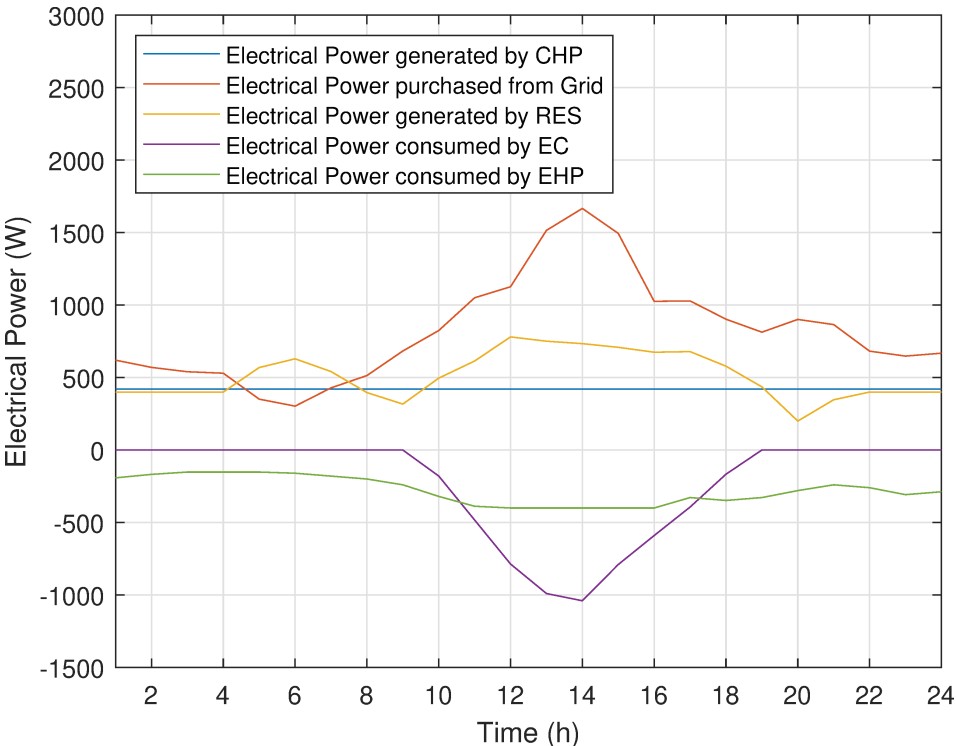

**Fig 12. Electrical power of SREH in Case Study-III.**

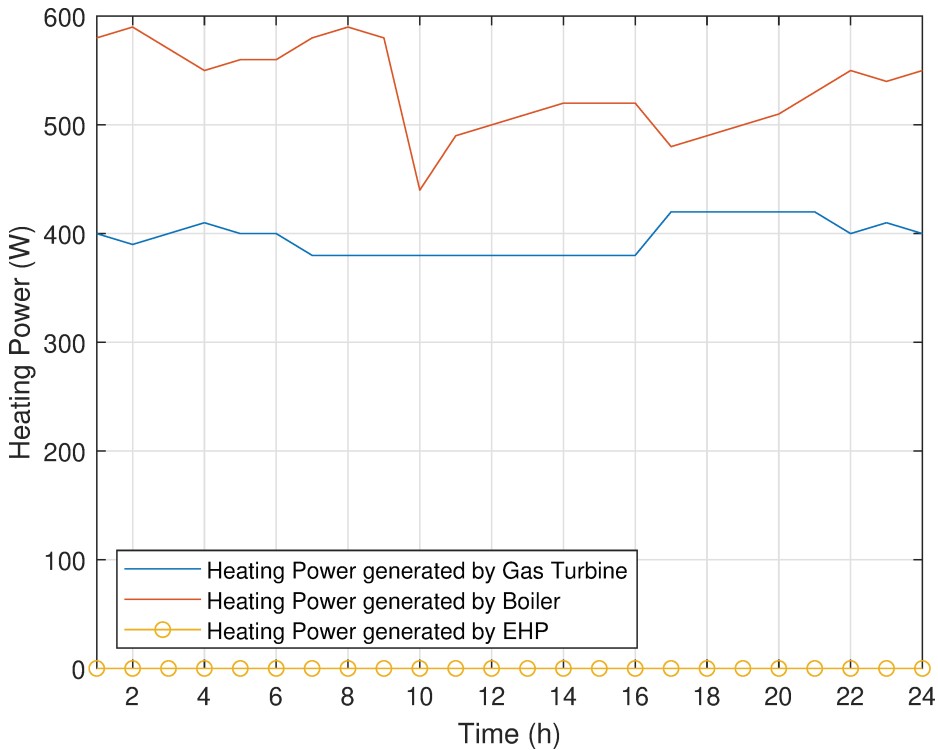

**Fig 13. Heating power of SREH in Case Study-III.**

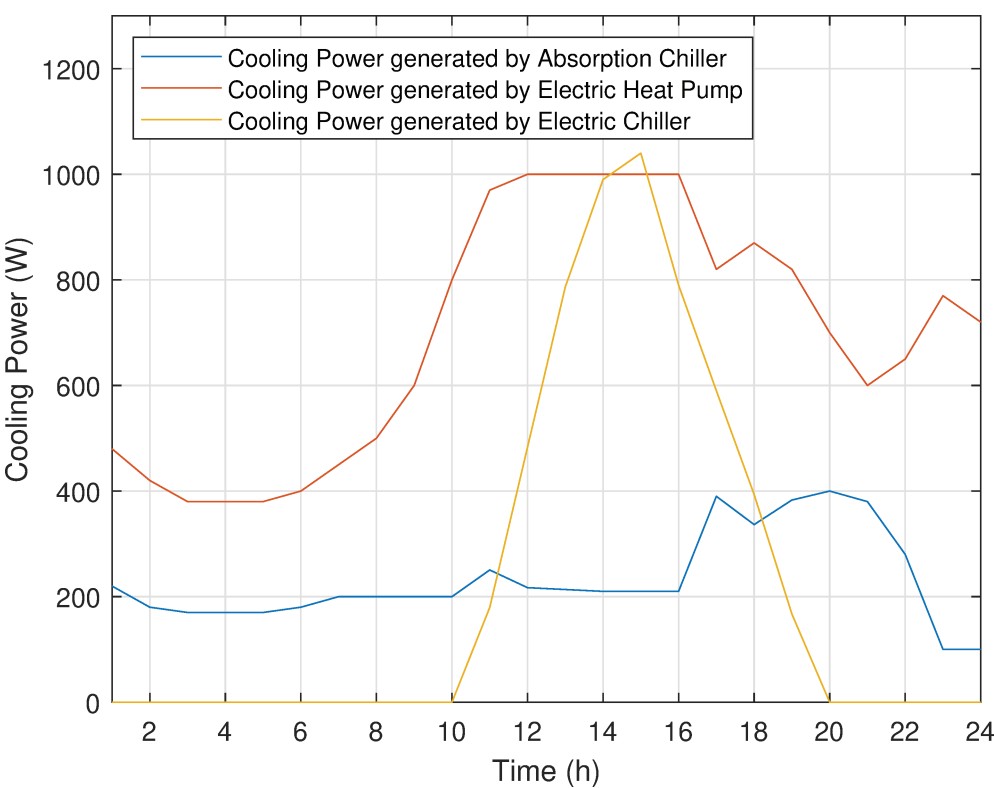

**Fig 14. Cooling power of SREH in Case Study-III.**

## 6.4 Case Study-IV

In Case Study IV, the DSM technique is introduced to further optimize the operation of the REH. DSM involves actively managing the energy demand by shifting some loads from peak hours to off-peak hours, reducing the dependance on grid energy during high-cost periods. We assume that 10% of the loads are shiftable. These shiftable loads include non-essential or flexible energy-consuming appliances and systems that do not need to operate during peak hours. The DSM strategy redistributes a portion of the energy demand by delaying or advancing operations to occur during off-peak hours, when electricity and gas prices are lower. This reduces the dependency on purchased energy during peak hours when tariffs are higher. The ability to shift portions of the load from peak to off-peak hours decreases dependence on purchased electricity and gas during peak times, leading to further cost reductions. Fig. 15 illustrates the electricity and gas purchased from the respective grids in Case Study IV. This load shifting by DSM results in a further reduction of the operating costs of the REH to 7439.68 USD, reflecting an additional decrease of approximately 1.11%. The electric, heating, and cooling portion of REH in Case Study-IV is shown in Figs 16 to 18. The total operational cost reduction from Case Study II to Case Study IV is significant, highlighting the cumulative benefits of smart energy management techniques. Shifting some portions of the load also enhances the flexibility in charging and discharging of the PHEV battery, as demonstrated in Fig. 19. The application of DSM ensures that load is spread more evenly across the day, reducing peaks and taking advantage of off-peak energy rates. This case highlights the importance of DSM as a key strategy for reducing energy costs and improving the efficiency of renewable energy systems in REH.

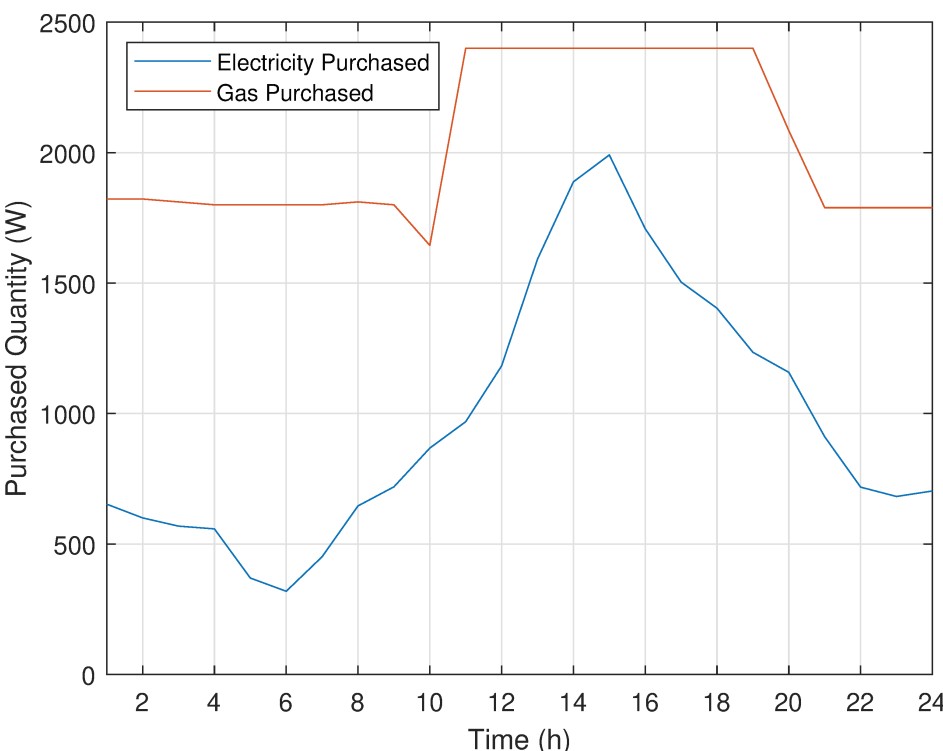

**Fig 15. Electricity and natural gas purchased from the grid in Case Study-IV.**

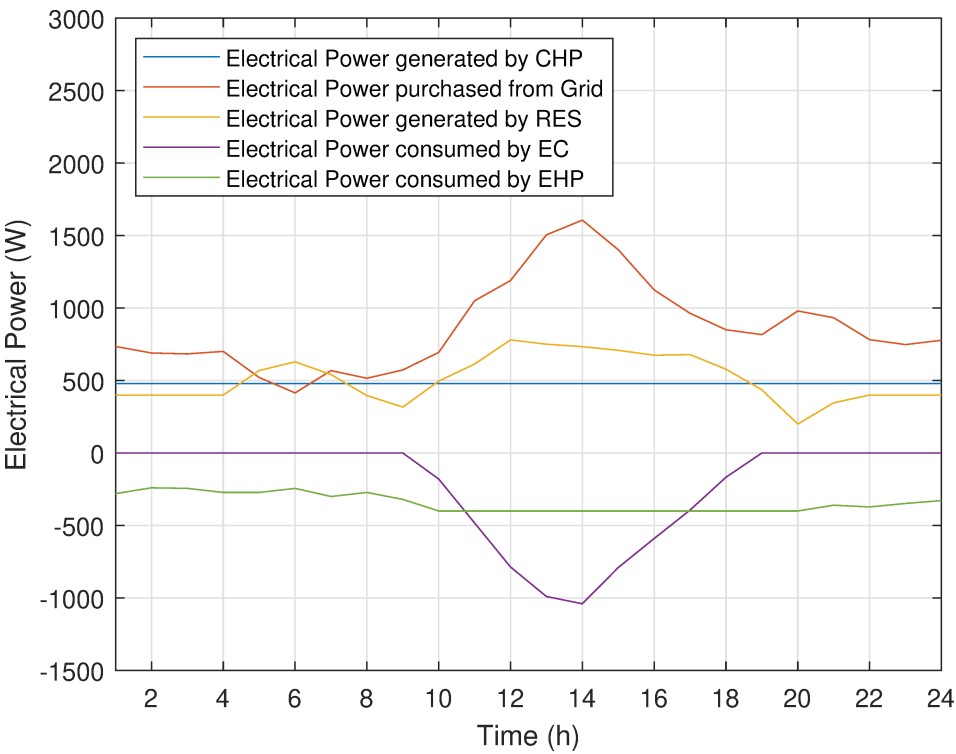

**Fig 16. Electrical power of SREH in Case Study-IV.**

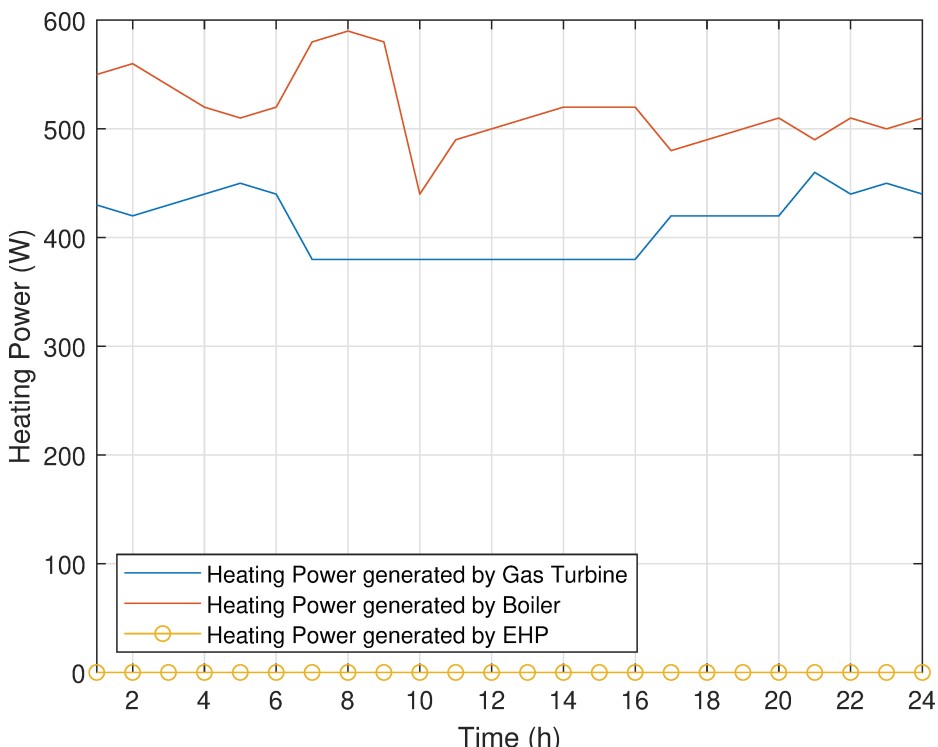

**Fig 17. Heating power of SREH in Case Study-IV.**

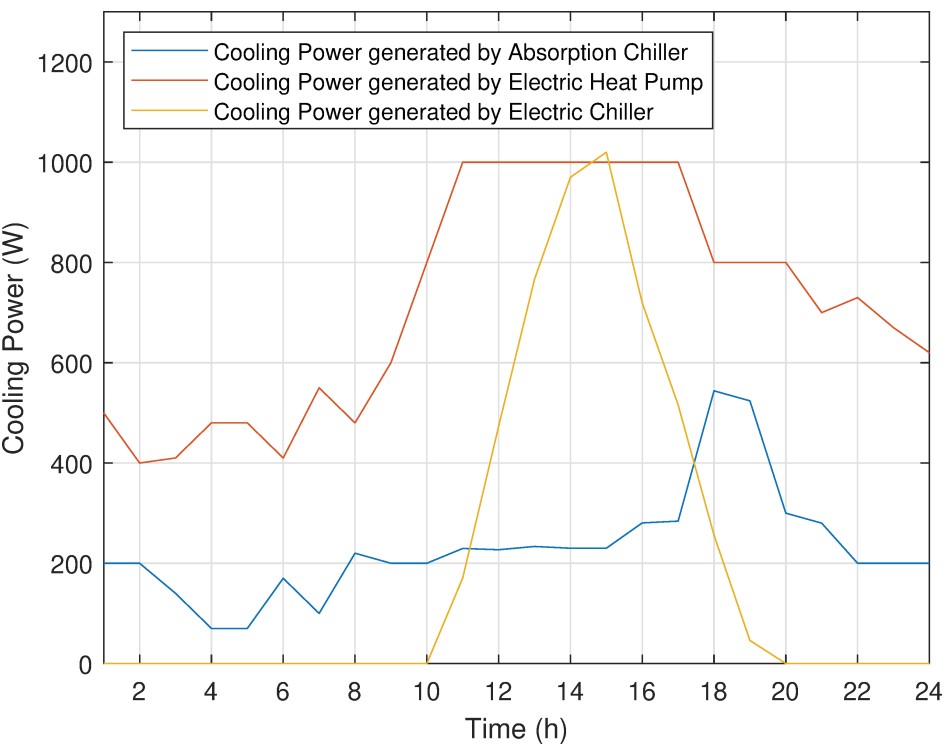

**Fig 18. Cooling power of SREH in Case Study-IV.**

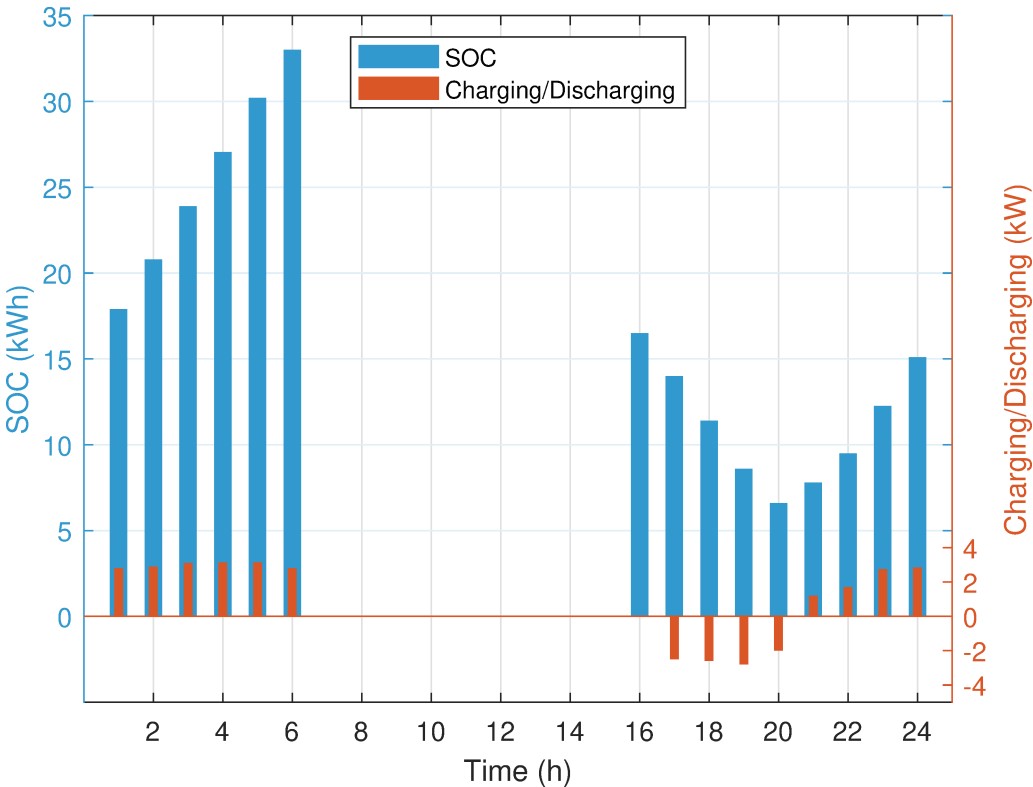

**Fig 19. SOC and charging/discharging of PHEV battery in Case Study-IV.**

## 6.5 Cases summary

The first case is a base case in which the total operational cost of REH is 7675.00 USD. The second case incorporates the PHEV without any smart charging mechanism, which increases the operational cost of REH by 0.63% due to the additional charging of PHEV. The third case involves the smart charging mechanism, which takes care of peak/off-peak hours and smartly charges/discharges the PHEV. The employed smart charging mechanism decreases the cost of SREH by 1.98% while also charging the PHEV and increases the reliability of REH as well. The fourth case involves DSM in addition to smart charging, which results in a further decrease in cost by 3.06%. A summary of these results is shown in Table 8.

## 6.6 Discussion

The first case is the base case, containing all the basic components of the REH, such as RES, CHP, AC, EC, a boiler, and electrical, heating, and cooling loads, but without the involvement of a PHEV. After a proper sensitivity analysis of key REH parameters, the sizes of different components are carefully selected, keeping practical systems in mind. The parameters of RES and central devices are optimally selected so that they are neither too large, which would result in high investment or operational costs, nor too small, making their impact negligible. After selecting the desired configuration, the REH is optimized for its operational cost over 24 hours. The second case involves incorporating a PHEV into the REH without any smart charging mechanism to analyze the impact of adding a PHEV to the REH. The unmanaged charging of the PHEV increases the operational cost of the REH, highlighting the need

**Table 8. Operational cost analysis of SREH under different case studies.**

| Case Study | PHEV Inclusion | Smart Charging of PHEV | Demand Side Management | Total Cost (USD) | % Change |
|---|---|---|---|---|---|
| I | - | - | - | 7675.00 | - |
| II | ✓ | - | - | 7723.70 | +0.63 |
| III | ✓ | ✓ | - | 7523.10 | -1.98 |
| IV | ✓ | ✓ | ✓ | 7439.68 | -3.06 |

for proper smart charging of the PHEV. In the third case, a smart charging mechanism is introduced that optimally charges/discharges the PHEV, taking into account the pricing of purchased electricity at any given time. The PHEV charging/discharging process is specified as a set of decision variables based on the pricing mechanism. This enables the PHEV to be charged when electricity prices are low and discharged when they are high. The flexibility of discharging the PHEV to obtain electrical energy when the system is under stress increases the stability and reliability of the REH, making it more robust and resilient. The fourth case involves DSM, which shifts some portion of the load when required. When there is increased load stress on the REH during peak hours, DSM makes it possible to optimally shift some shiftable loads to off-peak hours, reducing the need to purchase high-cost energy. This strategy not only increases the stability of the REH but also decreases its operational cost.

## 7 Conclusions

This article presents a stochastic MILP model of REH that integrates with RES in the presence of PHEV. Four case studies were conducted to assess the effectiveness of the proposed scheme. Firstly, the proposed scheme optimizes REH in the presence of RES, meeting its electrical demand through a combination of CHP, RES, and grid-purchased electricity. The results show that when the price of gas is less than the electricity price, the CHP unit generates more electrical energy from the gas to avoid the costly purchase of electricity. EHP converts the electrical energy into cooling power instead of heating energy due to the high demand for cooling energy in summer weather. Without any charging management, adding the PHEV to the REH setup causes the operating cost of REH to increase, rather than decrease, as it charges during peak hours. However, smart charging and discharging of the PHEV allow SREH to significantly reduce its operational expenses. Smart management of a PHEV battery enables the consumption of stored electrical energy during peak hours, eliminating the need to purchase it from the grid and recharging it during off-peak hours for the next day's trip. Finally, the DSM technique is applied to all types of loads to further reduce operational costs without compromising customer comfort. In terms of practical applications, the proposed model can be implemented in smart buildings, residential apartments, and societies to optimize energy efficiency and reduce reliance on expensive grid electricity. By implementing smart energy management strategies, real-world applications of this model can significantly reduce energy costs and carbon footprints for residential consumers. To make the overall energy infrastructure more resilient and cost-effective, urban planners and policymakers can apply these insights to build sustainable microgrids that seamlessly integrate RES, PHEVs, and ESS. This study has not considered the actual degradation of batteries in PHEVs and their long-term economic implications. Future research should aim to incorporate the impact of battery degradation and its replacement costs. Investigating the integration of emerging technologies, such as hydrogen storage and vehicle-to-grid interactions, could further improve the flexibility and sustainability of the REH.

## Author contributions

**Conceptualization:** Nouman Qamar.

**Data curation:** Nouman Qamar.

**Formal analysis:** Mohammed Alqahtani.

**Project administration:** Mohammed Alqahtani, Ijaz Ahmed.

**Supervision:** Muhammad Rehan, Ijaz Ahmed, Muhammad Khalid.

**Writing – original draft:** Nouman Qamar, Mohammed Alqahtani.

**Writing – review & editing:** Muhammad Rehan, Ijaz Ahmed, Muhammad Khalid.

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
