## [Decision Letter · Decision Letter 0]

PONE-D-25-06050Stochastic Optimization for Minimizing Operational Costs in Smart Hybrid Energy Networks Considering Electric VehiclePLOS ONE

Dear Dr. Khalid,

Thank you for submitting your manuscript to PLOS ONE. After careful consideration, we feel that it has merit but does not fully meet PLOS ONE’s publication criteria as it currently stands. Therefore, we invite you to submit a revised version of the manuscript that addresses the points raised during the review process.

**ACADEMIC EDITOR: please revise accordingly**==============================

We look forward to receiving your revised manuscript.

Kind regards,

Zhengmao Li

Academic Editor

PLOS ONE

Journal Requirements:

“The authors would like to express their profound gratitude to King Abdullah City for Atomic and Renewable Energy (K.A.CARE) for their financial support in accomplishing this work at KFUPM. The author MA would like to acknowledge the support by the Deanship of Scientific Research through King Khalid University, Saudi Arabia funded by the Large Group Research Project RGP2/392/45.”

“The authors would like to express their profound gratitude to King Abdullah City for Atomic and Renewable Energy (K.A.CARE) for their financial support in accomplishing this work at KFUPM. The author MA would like to acknowledge the support by the Deanship of Scientific Research through King Khalid University, Saudi Arabia funded by the Large Group Research Project RGP2/392/45.”

“The author(s) received no specific funding for this work”

Reviewers' comments:

Reviewer's Responses to Questions

**Comments to the Author**

1. Is the manuscript technically sound, and do the data support the conclusions?

Reviewer #1: Yes

Reviewer #2: Yes

2. Has the statistical analysis been performed appropriately and rigorously? 

Reviewer #1: Yes

Reviewer #2: Yes

3. Have the authors made all data underlying the findings in their manuscript fully available?

Reviewer #1: Yes

Reviewer #2: Yes

4. Is the manuscript presented in an intelligible fashion and written in standard English?

Reviewer #1: Yes

Reviewer #2: Yes

5. Review Comments to the Author

Reviewer #1: This paper proposes a data-driven distributionally robust optimization approach for integrated energy systems. While the paper is well-written, I still have the following comment:

1.It is recommended to delete the second column of the table, as all the literature includes RES, making it non-comparative.

2.The studied REH structure is relatively simple. It is suggested to extend the components or provide more details about the component models, such as modeling different types of transferable and non-transferable loads separately.

3.It is advisable to list and analyze the results for each case, as this would make the comparisons clearer and easier to understand.

4.It is recommended to expand the conclusion section to outline the limitations of this study and provide a more detailed discussion of future research needs.

5.The proposed optimization method has already been extensively researched, thus the contribution of the manuscript is insufficient. It is suggested to highlight the novelty of the proposed method and conduct a comparative analysis with existing methods.

6.What are the differences between the proposed intelligent charging and discharging strategy for PHEVs and that of a regular battery?

Reviewer #2: This paper focuses on the optimization of the operational costs of Residential Energy Hubs (REHs) in smart hybrid energy networks. The proposed stochastic model and related strategies demonstrate certain innovativeness. However, there are some areas that need to be improved, and my comments are as follows:

1. The assumptions for handling the uncertainties of Renewable Energy Sources (RES) and Plug - in Hybrid Electric Vehicles (PHEVs) in the paper are relatively simplistic. It is advisable to introduce more complex stochastic processes or probability distribution models to enhance the accuracy and practicality of the model.

2. Sensitivity analysis should be added to clarify the impact degree of each parameter on the results, providing more targeted guidance for the optimization of practical systems.

3. Although the four case studies compare the operation of REHs under different scenarios, the experimental design is not comprehensive enough. It is recommended to add case analyses of different seasons and regions to make the research results more universal.

4. The analysis of the experimental results is rather superficial, mainly staying at the level of data comparison. It is necessary to deeply explore the reasons behind the data, such as the specific mechanisms by which the smart charging mechanism and the Demand Response Program (DRP) reduce costs, as well as their impacts on the stability and reliability of the energy system.

5. The research results are not closely integrated with practical applications. It is recommended to discuss this in the conclusion.

6. It is advisable to update the literature review, supplement the latest research progress, clarify the relationship between this research and the cutting - edge achievements, and highlight the innovativeness and necessity of the research. It is recommended to supplement the following literature (not mine): 10.1109/TCSI.2024.3523339; 10.1109/TII.2024.3495785.

6. PLOS authors have the option to publish the peer review history of their article (what does this mean?). If published, this will include your full peer review and any attached files.

Reviewer #1: No

Reviewer #2: No

---

## [Decision Letter · Decision Letter 1]

Stochastic Optimization for Minimizing Operational Costs in Smart Hybrid Energy Networks Considering Electric Vehicle

PONE-D-25-06050R1

Dear Dr. Khalid,

We’re pleased to inform you that your manuscript has been judged scientifically suitable for publication and will be formally accepted for publication once it meets all outstanding technical requirements.

Kind regards,

Zhengmao Li

Academic Editor

PLOS ONE

Additional Editor Comments (optional):

congrats

Reviewers' comments:

Reviewer's Responses to Questions

**Comments to the Author**

1. If the authors have adequately addressed your comments raised in a previous round of review and you feel that this manuscript is now acceptable for publication, you may indicate that here to bypass the “Comments to the Author” section, enter your conflict of interest statement in the “Confidential to Editor” section, and submit your "Accept" recommendation.

Reviewer #1: All comments have been addressed

Reviewer #2: All comments have been addressed

2. Is the manuscript technically sound, and do the data support the conclusions?

Reviewer #1: Yes

Reviewer #2: (No Response)

3. Has the statistical analysis been performed appropriately and rigorously? 

Reviewer #1: Yes

Reviewer #2: (No Response)

4. Have the authors made all data underlying the findings in their manuscript fully available?

Reviewer #1: No

Reviewer #2: (No Response)

5. Is the manuscript presented in an intelligible fashion and written in standard English?

Reviewer #1: Yes

Reviewer #2: (No Response)

6. Review Comments to the Author

Reviewer #1: The authors have addressed all reviewers' comments, and the revised paper is ready for publication.

Reviewer #2: (No Response)

7. PLOS authors have the option to publish the peer review history of their article (what does this mean?). If published, this will include your full peer review and any attached files.

Reviewer #1: No

Reviewer #2: No

---

## [Editor Report · Acceptance letter]

PONE-D-25-06050R1

PLOS ONE

Dear Dr. Khalid,

I'm pleased to inform you that your manuscript has been deemed suitable for publication in PLOS ONE. Congratulations! Your manuscript is now being handed over to our production team.

Kind regards,

on behalf of

Dr Zhengmao Li

Academic Editor

PLOS ONE